# Effects of Different Nitrogen Levels on Lignocellulolytic Enzyme Production and Gene Expression under Straw-State Cultivation in *Stropharia rugosoannulata*

**DOI:** 10.3390/ijms241210089

**Published:** 2023-06-13

**Authors:** Jinjing Zhang, Xinyi Zhuo, Qian Wang, Hao Ji, Hui Chen, Haibo Hao

**Affiliations:** 1National Research Center for Edible Fungi Biotechnology and Engineering, Key Laboratory of Applied Mycological Resources and Utilization, Ministry of Agriculture, Shanghai Key Laboratory of Agricultural Genetics and Breeding, Institute of Edible Fungi, Shanghai Academy of Agricultural Sciences, Shanghai 201403, China; hf.zjj6688@163.com (J.Z.); 18742592498@163.com (X.Z.); wq-15309@163.com (Q.W.); 15256776956@163.com (H.J.); 2College of Life Science, Nanjing Agricultural University, Nanjing 210095, China; 3College of Food Sciences and Technology, Shanghai Ocean University, Shanghai 200090, China; 4State Key Laboratory of Genetic Engineering and Fudan Center for Genetic Diversity and Designing Agriculture, Institute of Plant Biology, School of Life Sciences, Fudan University, Shanghai 200438, China

**Keywords:** *Stropharia rugosoannulata*, nitrogen metabolism, cellulase metabolism, CAZymes, MAPK signaling pathway

## Abstract

*Stropharia rugosoannulata* has been used in environmental engineering to degrade straw in China. The nitrogen and carbon metabolisms are the most important factors affecting mushroom growth, and the aim of this study was to understand the effects of different nitrogen levels on carbon metabolism in *S. rugosoannulata* using transcriptome analysis. The mycelia were highly branched and elongated rapidly in A3 (1.37% nitrogen). GO and KEGG enrichment analyses revealed that the differentially expressed genes (DEGs) were mainly involved in starch and sucrose metabolism; nitrogen metabolism; glycine, serine and threonine metabolism; the MAPK signaling pathway; hydrolase activity on glycosyl bonds; and hemicellulose metabolic processes. The activities of nitrogen metabolic enzymes were highest in A1 (0.39% nitrogen) during the three nitrogen levels (A1, A2 and A3). However, the activities of cellulose enzymes were highest in A3, while the hemicellulase xylanase activity was highest in A1. The DEGs associated with CAZymes, starch and sucrose metabolism and the MAPK signaling pathway were also most highly expressed in A3. These results suggested that increased nitrogen levels can upregulate carbon metabolism in *S. rugosoannulata*. This study could increase knowledge of the lignocellulose bioconversion pathways and improve biodegradation efficiency in Basidiomycetes.

## 1. Introduction

*Stropharia rugosoannulata* Farl. ex Murrill, commonly known as the wine-cap Stropharia mushroom or king *Stropharia*, is an edible fungus with high medicinal and nutritional value [1]. As an environmentally friendly fungus, *S. rugosoannulata* is mainly cultivated on straw and does not consume forest resources [2]. Its growth substrates are mainly agricultural and forestry wastes, such as rice straw, wheat straw, corn straw and various dead tree branches, and it is cultivated in fields, woodlands and simple greenhouses. In China, *S. rugosoannulata* has been widely cultivated, not only because it can use a variety of agricultural waste resources to grow and produce a certain economic value, but also because it is an important environmental engineering strain to treat agricultural and forestry wastes [2]. The main carbon source of agricultural wastes is lignocellulose, which consists of three main polymers: cellulose, hemicellulose and lignin [3]. Nitrogen is another important factor that might affect lignocellulosic degradation efficiency in fungi [4]. However, the mechanism underlying the regulatory effect of nitrogen on carbon metabolism in *S. rugosoannulata* is still unclear.

Nutrient elements play important roles in the growth and development of organisms. Nitrogen is a vital nutrient for microorganisms because it plays an essential role in microbial cell development and helps sustain a fast growth rate [5]. Generally, fungi can uptake a variety of nitrogen sources, such as nitrate, ammonium and urea, which involve multiple regulatory genes [6]. The major enzymes involved in nitrogen metabolism are nitrate reductase (NR), nitrite reductase (NiR), glutamine synthetase (GS) and glutamine-2-oxoglutarate aminotransferase (GOGAT, also known as glutamate synthase). NR and NiR mediate the initiation of inorganic nitrogen utilization, and the GS/GOGAT cycle converts inorganic nitrogen into organic nitrogen, thus playing a vital role in nitrogen assimilation [7]. Asparagine synthetase (AS) and glutamate dehydrogenase (GDH) are also involved in nitrogen metabolism. AS catalyzes the transfer of an amino group from glutamine to aspartate to form glutamate and asparagine, which is crucial for glutamine metabolism in Saccharomyces cerevisiae [8]. In yeast, in addition to the GS/GOGAT pathway, the GDH pathway is a conserved mechanistic pathway for glutamate synthesis and nitrogen assimilation, wherein glutamate is synthesized using α-ketoglutarate and ammonium through the NADP-linked action of GDH [9,10].

Carbon is another important factor in balancing biomass for fungal growth. In basidiomycetes, the main forms of carbon are lignin, hemicellulose and cellulose; the lignin and hemicellulose molecules act as a barrier blocking the chemical interaction of cellulose and cellulase enzymes, thereby slowing enzymatic hydrolysis [11,12]. Cellulases are a family of hydrolytic enzymes that mainly comprise endoglucanases, exoglucanases, cellobiohydrolases and β-glucosidases. Generally, cellulase enzymes such as endoglucanase, cellobiohydrolase and β-glucosidase belong to a diverse range of glycoside hydrolase (GH) families that are classified in the Carbohydrate-Active EnZymes (CAZy) database [13]. In white-rot basidiomycetes, carbohydrate-active enzymes (CAZymes) primarily consist of GHs, carbohydrate esterases (CEs), carbohydrate-binding modules (CBMs), auxiliary activity enzymes (AAs) and polysaccharide lyases (PLs) [14].

The concentrations of nitrogen sources are important factors that have been shown to significantly influence the regulation of cellulase enzymes in white-rot basidiomycetes [15]. Both excess nitrogen and lack of nitrogen in the substrate may be limiting factors for the growth rate [16,17]. Nitrogen sources are usually added to growth media in appropriate amounts to ensure a proper nitrogen balance [18,19], which affects enzyme activities and gene expression, especially those of cellulases and xylanases [20,21,22]. The nature and concentration of nitrogen sources are important nutritional factors that have been shown to significantly influence the regulation of cellulase enzymes in white-rot basidiomycetes [23]. In Trichoderma reesei, while the nitrogen regulator was downregulated, cellulase production was reduced, and the expression levels of cellulase genes were also decreased [24]. In contrast, nitrogen has been indicated to repress the expression of lignolytic enzymes in white-rot basidiomycetes [25].

The objective of this study was to reveal how different nitrogen levels affect nitrogen and carbon metabolism-related enzymes of *S. rugosoannulata* by using corn straw as the main substrate. Moreover, RNA-seq transcriptome profiling was applied to analyze the expression levels of genes involved in substrate degradation to reveal the intrinsic biological mechanisms. Overall, the findings improve our knowledge of lignocellulose degradation by *S. rugosoannulata* based on a corn straw solid medium, which will be beneficial for straw usage.

## 2. Results

### 2.1. Nitrogen Levels Regulated the Mycelial Growth of S. rugosoannulata

Mycelial growth and mycelial branching were markedly affected by the three different nitrogen levels (Figure 1A). To quantify the effects of nitrogen levels on the vegetative growth of *S. rugosoannulata*, the mycelial growth rate of *S. rugosoannulata* under A1, A2 and A3 was measured after inoculation for 10 days. As shown in Figure 1A, mycelia grew more quickly in A3 than in A1 and A2 (Figure 1A). The nitrogen levels in the three solid media were 0.39%, 0.68% and 1.37%, respectively (Figure 1B). Mycelial growth increased at higher nitrogen levels and was significantly higher in A3 than in A1 and A2 (Figure 1C; *p* < 0.05). In addition, the mycelia were highly branched in A3 (Figure 1A), and the mycelial distance between two branches increased significantly in A3 compared with A1 or A2 (*p* < 0.05; Figure 1D). These results suggested that higher nitrogen levels induce higher mycelial branching, which might induce nutrient absorption and increase the mycelial growth of *S. rugosoannulata*.

### 2.2. Global Transcriptomic Analysis of S. rugosoannulata and Identification of Differentially Expressed Genes (DEGs)

To examine the differences among the three treatments, a PCA cluster diagram was constructed among nine samples based on the FPKM values (Figure 2A). The contributions of sample differences were 79.45% and 15.67% for PC1 and PC2, respectively, and the difference between A1 and A2 was less than that between A1 and A3 or between A2 and A3. To describe the gene expression patterns under three different nitrogen levels, nine libraries were constructed for *S. rugosoannulata*. A total of 383.14 million raw reads were generated via Illumina sequencing. After cleanup and quality control, 360.96 million clean reads were obtained, and the Q30 value of the base ratio was higher than 93.01% (Appendix A). In addition, 88.97–92.19% of the reads could be mapped to the *S. rugosoannulata* genome (Appendix A). The obtained RNA sequences were assembled by using the sequence clustering software Trinity, with 11495 genes annotated via the nonredundant (NR), STRING gene, GO, Clusters of Orthologous Groups (COGs), KEGG, and Swiss-Prot databases with E-values of 10–5.

Based on DEG analysis of the transcriptomes of *S. rugosoannulata* cultured under the three nitrogen levels, a total of 1207 DEGs were identified between the A1 and A2 samples, of which 461 DEGs were upregulated and 746 DEGs were downregulated in A2 (Figure 2B). A total of 1505 DEGs were identified between the A2 and A3 samples, of which 642 DEGs were upregulated and 862 DEGs were downregulated in A2 (Figure 2B). The largest number of DEGs was identified between the A3 and A1 samples, and the number of downregulated DEGs (1378) was greater than that of upregulated DEGs (663) in A1 (Figure 2B). A Venn diagram revealed 201 overlapping DEGs among the three comparisons (Figure 2C). Analysis of the gene expression patterns of these DEGs showed that some DEGs were significantly upregulated in A3, while more genes were downregulated in the A1 and A2 samples (Figure 2D). The gene expression patterns were similar between the A1 and A2 samples (Figure 2D). These results reveal that more genes were induced under higher nitrogen levels to enable proper metabolism in *S. rugosoannulata*.

### 2.3. Kyoto Encyclopedia of Genes and Genomes (KEGG) Pathway and Gene Ontology (GO) Enrichment Analyses of the Differentially Expressed Genes Were Used

To analyze the DEG expression patterns, the DEGs were divided into six different clusters. As shown in Figure 3A, the DEG expression levels of cluster 1 and cluster 4 were upregulated from the A1 to A3 samples, which suggested that the genes of the two clusters had important roles in nitrogen metabolism in *S. rugosoannulata*. Furthermore, the DEGs of the two clusters were analyzed via GO and KEGG enrichment analysis (Appendix A). The main enriched GO terms included carbohydrate derivative transport (70), cell periphery (306), organic substance transport (322), response to chemical (343) and hydrolase activity, and acting on glycosyl bonds (72) (Appendix A). The main enriched KEGG pathways included protein processing in endoplasmic reticulum (ko04141), starch and sucrose metabolism (ko00500), the MAPK signaling pathway–yeast (ko04011) and amino sugar and nucleotide sugar metabolism (ko00520) (Appendix A).

In addition, all of the DEGs were mapped to the KEGG database, and the top 20 significant pathways were examined to determine their functions. As shown in Figure 3 and Appendix A, the comparison between A1 and A2 revealed that the main enriched pathways were involved in starch and sucrose metabolism (ko00500); glycine, serine and threonine metabolism (ko00260); amino sugar and nucleotide sugar metabolism (ko00520) and pyruvate metabolism (ko00620) (Figure 3B). These findings suggested that the transition from A1 to A2 primarily involved the degradation of nitrogen and carbon resources in *S. rugosoannulata*. In the comparison between A2 and A3, the DEGs were mainly involved in starch and sucrose metabolism (ko00500); glycine, serine and threonine metabolism (ko00260); nucleotide excision repair (ko03420); arginine and proline metabolism (ko00330); DNA replication (ko03030) and nitrogen metabolism (ko00910) (Figure 3C). These results suggested that carbon metabolism was related to nitrogen metabolism in *S. rugosoannulata* in response to different nitrogen levels.

Based on the GO functional classification, all DEGs were classified into three different categories, and the top 20 terms for each comparison are shown in Appendix A and Appendix A. Transporter activity was increased in the comparison between A1 and A2 samples; the enriched terms included drug transport (69), drug transmembrane transport (67), secondary active transmembrane transporter activity (67), nucleotide transport (52) and carbohydrate derivative transmembrane transporter activity (67). These findings suggested that carbohydrate degradation activity was regulated by different nitrogen levels in *S. rugosoannulata*. In the A2 vs. A3 comparison, the significantly enriched terms included hydrolase activity on glycosyl bonds (45), hydrolase activity on O-glycosyl (44), an extracellular region (56), a hemicellulose metabolic process (15) and a polysaccharide catabolic process (31). These findings indicated that the substrate degradation activity was active in the transition from A2 to A3. Overall, growth at the three different nitrogen levels involved multiple pathways, such as nitrogen degradation, carbohydrate degradation and MAPK signaling regulation.

### 2.4. Analysis of Nitrogen Metabolism Genes at the Three Nitrogen Levels

Nitrogen is an essential element in organisms, as it is a fundamental component of proteins, amino acids and nucleic acids. KEGG enrichment analysis revealed that the nitrogen metabolism pathway (ko00910) was enriched in the two comparisons (A1 vs. A2 and A2 vs. A3) (Figure 3B,C). Therefore, the genes involved in regulating nitrogen metabolism in *S. rugosoannulata* were studied. As shown in Figure 4A, seven genes had significantly different expression levels in the three treatments, including NR, the NAD-specific glutamate dehydrogenase 1 (GDH1) gene, the cyanate hydratase 1 (CH1) gene, carbonic anhydrase (CA), the nitronate monooxygenase (NM) gene, the acetamidase (AM) gene and the glutamate dehydrogenase 2 (GDH2) gene. In the A1 vs. A2 comparison, six genes (Nr, Gdh1, Ca, Ch1, Am and Nm) were upregulated in A1, and only one gene (Gdh2) was upregulated in A2. In the A2 vs. A3 comparison, four genes (Am, Nr, Gdh1 and Ch1) were upregulated in A2, while three genes (Nm, Gdh2 and Ca) were upregulated in A3. These results suggest that these genes had different expression patterns in response to different nitrogen levels in *S. rugosoannulata*.

NR, NiR, GDH, GS, GOGAT and AS are the major enzymes in nitrogen metabolism. The activities of these enzymes, which are involved in both nitrogen metabolism and assimilation, were detected in the samples from the three nitrogen levels. As shown in Figure 4B,C, the activities of NR and NiR were significantly induced in A1 (*p* < 0.05) and showed no significant differences between the A2 and A3 samples (*p* > 0.05), which suggested that nitrogen assimilation was more active in A1 than the other two samples. In addition, the activity of GOGAT (Figure 4D) was significantly higher in the A1 sample than in the A2 and A3 samples (*p* < 0.05), and the activity in the A3 sample was significantly higher than that in the A2 sample (*p* < 0.05). The activity of GDH was significantly higher in the A1 sample than in the A2 and A3 samples (*p* < 0.05), but the A2 and A3 samples showed no significant differences in GDH activity (Figure 4E; *p* < 0.05). In addition, the AS activity was consistent with the trend in GOGAT activity, showing the highest level in A1, and A2 and A3 showed no significant differences (Figure 4F). The GS activity was consistent with the GDH activity; A1 had the highest activity, and A2 and A3 showed no significant differences (Figure 4G). These results suggested that nitrogen metabolism and assimilation were the highest in A1, suggesting that an increase in the nitrogen content inhibits the activity of nitrogen metabolism enzymes.

Notably, the glycine, serine and threonine metabolism pathways were significantly enriched in all the comparisons. Twenty-three genes were differentially expressed, including 13 GMC oxidoreductase (GMC) genes, two pyranose dehydrogenase 3 (PYD) genes, two amine oxidase (Amo) genes, one glyoxylate reductase (GLR) gene, one cystathionine gamma-lase (CYG) gene, one phosphoglycerate mutase (PHM) gene, one aryl-alcohol oxidase (AAO) gene and one 5-aminolevulinate synthase (AMS) gene (Table 1). Most of these genes were significantly upregulated in A1, including eight GMC genes, CYG, AAO, AMS and two PYD genes. In A2, only three genes were significantly upregulated, including two GMC genes and one AMO gene. In A3, seven genes were significantly upregulated, including three GMC genes, two Amo genes, one GLR gene and one PHM gene. These results suggested that the expression of genes in the glycine, serine and threonine metabolism pathways was regulated by different nitrogen levels and that the glycine, serine and threonine metabolism pathways might be important pathways involved in the degradation of amino acids in *S. rugosoannulata*.

### 2.5. Analysis of Carbon Metabolism Genes at the Three Nitrogen Levels

Nitrogen metabolism requires sufficient energy and carbon skeletons, which can be provided by carbohydrates. CAZymes are a large class of very important enzymes divided into six types: GHs, CEs, CBMs, AAs, glycosyl transferases (GTs) and PLs. In this study, the expression levels of carbohydrate enzyme genes were analyzed at the three nitrogen levels (Figure 5A). In A1 vs. A2, 52 genes were significantly upregulated, and 23 genes were significantly downregulated in A2 (*p* < 0.05), with 11 genes showing no significant differences between the A1 and A2 samples (*p* > 0.05). In the A2 vs. A3 comparison, 79 genes were significantly upregulated, and 32 genes were significantly downregulated in A3 (*p* < 0.05), with 16 genes showing no significant differences between the A2 and A3 samples (*p* > 0.05). Furthermore, 51 genes were specifically expressed in A1 vs. A2, 93 genes were specifically expressed in A2 vs. A3 and 36 genes overlapped between the two comparisons (Figure 5B). The gene expression heatmap of the 36 genes was analyzed, which included 14 GH genes, 10 AA genes, 10 CE genes, 1 CBM gene and 1 GT gene. Eight genes were significantly upregulated in the A1 sample, including GH25 (DQGG002590), GH109 (DQGG9887), AA3 (DQGG008585, DQGG3062), CE10 (DQGG9290, DQGG002082, DQGG001136) and CE16 (DQGG003326). Nine genes were significantly upregulated in A2, including GH16 (DQGG006143), GH79 (DQGG003958), GH16 (DQGG010262), GH10 (DQGG007454), GH5 (DQGG006934), AA2 (DQGG003998), AA3 (DQGG002955, DQGG3527) and CBM1 (DQGG002944). Nineteen genes were significantly upregulated in A3, including AA2(DQGG000504, DQGG001116, 003989), AA3(DQGG005582), GH28(DQGG004628), GH7(DQGG002368), GH127(DQGG002173), GH10(DQGG010271), GH5(DQGG009707), GH53(DQGG008921), GH11(DQGG004511), CE4(DQGG003952), CE10(DQGG010939), CE16(DQGG004257), CE15(DQGG000402) and CE5(DQGG011314). The expression pattern of carbohydrate enzyme genes was analyzed at the three nitrogen levels (Appendix A) and was consistent with that of these 36 genes (Figure 5B). These results indicated that many carbohydrate enzyme genes (AAs, GHs, CEs and GTs) were upregulated by the high nitrogen level in *S. rugosoannulata*.

KEGG enrichment analysis revealed that starch and sucrose metabolism pathways were significantly enriched at the three nitrogen levels. Therefore, the genes involved in carbon metabolism should be detected at different nitrogen levels in *S. rugosoannulata*. As shown in Figure 5C, most genes associated with starch and sucrose metabolism had the highest expression levels in the A3 samples, including GT20 (DQGG003175), GA (DQGG000349), GH13 (DQGG007170), GH15 (DQGG000348), GH5 (DQGG009707), GH1 (DQGG009756), Cx (DQGG), GH31 (DQGG002694), GH31 (DQGG002694), GPU1 (DQGG004894), GPU2 (DQGG004895), GH5 (DQGG003769) and β-GC (DQGG007674). The gene expression levels of PH (DQGG000609), GH13 (DQGG007170), GH74 (DQGG002944) and β-GA (DQGG006934) were the highest in A2. However, the significant DEGs related to starch and sucrose metabolism had the lowest expression levels in A1. These results were consistent with the analyses of carbohydrate enzyme genes, indicating that carbon metabolism was induced by increased nitrogen levels in *S. rugosoannulata*. In addition, the activities of cellulases (Cx, β-GC and β-GA) and ACX were also detected. The results showed that the activities of β-GA (Figure 5D), Cx (Figure 5E) and β-GC (Figure 5F) were all the highest in A3, while that of ACX was the highest in A1 (Figure 5G). These results were in accordance with those regarding the activity of nitrogen metabolism enzymes. These results suggested that cellulases and ACX played important roles in providing energy and carbon skeletons for nitrogen metabolism in *S. rugosoannulata* and that the activity of cellulases was increased by higher nitrogen levels.

### 2.6. Analysis of the Mitogen-Activated Protein Kinase (MAPK) Signaling Pathway at the Three Nitrogen Levels

MAPK signaling pathways are critical for the ability of eukaryotic cells to sense and respond to changes in nutrition. KEGG enrichment analysis revealed that MAPK signaling pathway genes were enriched. In A1 vs. A2 (Figure 6A), three genes were significantly differentially expressed, including pheromone B beta 1 receptor (DQGG001215), pheromone B alpha 1 receptor (DQGG001226) and protein tyrosine kinase (DQGG011111). In A2 vs. A3 (Figure 6B), 13 genes were significantly differentially expressed, and most of these DEGs were upregulated in A3. Furthermore, the heatmap of these DEGs is shown in Figure 6C and shows that most of the MAPK signaling pathway genes were upregulated in A3. In addition, three genes (Mapk, Sho1 and Pba) were randomly selected for qRT–PCR, and the qRT–PCR results were consistent with the transcript expression patterns (Figure 6D). These results suggested that the MAPK signaling pathway was induced by higher nitrogen levels in *S. rugosoannulata*.

### 2.7. Validation of Transcriptomic Data by qRT–PCR

To validate the RNA-seq findings used to draw conclusions in this study, qRT–PCR was used. Based on the gene expression patterns, typical genes involved in carbon and nitrogen metabolism among these DEGs were selected for qRT–PCR analysis. Six nitrogen metabolism genes (Figure 7A) and four carbon metabolism enzyme genes (Figure 7B) were identified: NR (DQGG006945), NiR (DQGG001090), AS (DQGG006918), GS (DGQQ001244), GOGAT (DQGG006969), GDH (DQGG009462), Cx (DQGG008787), β-GA (DQGG006934), β-GC (DQGG009756) and ACX (DQGG002951). Most of these enzyme-encoding genes were significantly upregulated in A1 (*p* < 0.05), similarly to the transcript expression pattern. In addition, the expression patterns of nine CAZyme genes were verified (Figure 7C). Most of the genes were significantly upregulated in A1 (*p* < 0.05), similarly to the transcript expression pattern. However, there were some differences in mRNA and protein levels between these genes, indicating that there might exist posttranscriptional control mechanisms for these proteins.

## 3. Discussion

The aim of this study was to study the mechanism by which different nitrogen levels regulate straw degradation by *S. rugosoannulata*. In this study, three nitrogen levels were designed to culture *S. rugosoannulata*, and the mycelia were found to be highly branched and elongated quickly in A3. The lack of nitrogen in the substrate affected the mycelial growth of *S. rugosoannulata*, in accordance with the results of Mantovani et al. [16]. In addition, the activity of cellulase was higher in A3 than in A1 or A2, but nitrogen metabolism enzyme activities were inhibited in A3. Furthermore, the expression of most genes in the CAZyme and starch and sucrose metabolic pathways was upregulated in A3. These results suggested that lignocellulose degradation was regulated by higher nitrogen levels in *S. rugosoannulata*.

Some studies have proven that both the nature and concentration of nitrogen sources are powerful nutrition factors regulating lignocellulolytic enzyme production by wood-rotting basidiomycetes [5,23]. In fungi, nitrogen regulates lignocellulolytic enzyme production and gene expression, including those of cellulases and ACXs [17,18,19,21,22]. In white-rot fungi, lignocellulolytic enzyme production exhibited different responses to nitrogen; only the xylanase activity of Pleurotus dryinus IBB 903 and the laccase activity of Lentinus edodes IBB 363 were increased, and the manganese peroxidase activities of four tested fungi were repressed [19]. Wei et al. [26] demonstrated that the activities of cellulase, xylanase, manganese peroxidase, lignin peroxidase and laccase were increased by adding urea as a nitrogen source for Actinomycetes. In this study, the activity of cellulases (Cx, β-GC and β-GA) and ACX was also measured, as nitrogen sources also affect the expression of the genes encoding these enzymes [21,22]. The results showed that the hemicellulase ACX had the highest activity in A1, consistent with the activity of nitrogen metabolism-related enzymes. In contrast, cellulase enzymes (Cx, β-GC and β-GA) had the highest activity in A3, indicating that cellulase and hemicellulase have different responses to different nitrogen levels in *S. rugosoannulata*.

Specific side groups of amino acids regulate the synthesis of various extracellular enzymes. Cristica et al. [15] reported that the addition of glutamic acid and asparagine to a growth medium can increase cellulase and β-ACX activity in the filamentous fungus Trichoderma reesei QM-9414, while the addition of methionine decreases enzyme activities. However, glutamine displayed the strongest inhibitory effect on cellulase production in *T. reesei* [27]. In Aspergillus fumigatus Z5, most cellulose-degrading enzymes, including cellulase, hemicellulases and glycoside hydrolase, are largely not regulated by cysteine [4]. These results indicated that amino acids might be involved in the regulation of lignocellulase production and that different amino acids might be differentially regulated in fungi. Similar to cellulase activity, most genes of the glycine, serine and threonine metabolism pathways were upregulated in A3. These results suggested that the glycine, serine and threonine metabolism pathways might be important pathways that regulates cellulase production under higher nitrogen levels of *S. rugosoannulata*.

Nitrogen metabolism links carbon metabolism through the GS/GOGAT cycle [28], which might be regulated by adding nitrogen sources. In Ganoderma lucidum, carbon metabolism pathways, such as glycolysis reactions and the TCA cycle, were also regulated by different nitrogen levels [29]. In Lentinula edodes, the starch and sucrose metabolism pathways were significantly enriched in a higher nitrogen content substrate [30]. In this study, most DEGs involved in the starch and sucrose metabolism pathway were significantly upregulated by higher nitrogen levels (A2 or A3). In the cellulose depolymerization process, the first important event is th random hydrolysis of β-1,4-glycosidic linkages by endoglucanases; then, the β-glucosidases convert cellobiose, the primary product of the endo- and exoglucanase mixture, to glucose [31]. In addition, β-1,3-glucanase (β-GA), endo-β-1,4-glucanase (Cx) and β-glucosidase (β-GC) had transcriptomic results consistent with the enzyme activity results, which suggested that carbon metabolism was upregulated at higher nitrogen levels (A2 and A3) in *S. rugosoannulata*.

Cellulases are the major members of the GH family that catalyze the hydrolysis of β-1,4-glycosidic bonds of cellulose to glucose [32,33], and these enzymes belong to the CAZyme family. Basidiomycetes can efficiently degrade lignocellulosic biomass, especially that derived from plants, because of their diverse CAZymes [13,34]. In this study, DEG analysis revealed that a series of genes associated with cellulase (GH5, GH7) and ACX (GH10) had the highest levels in A2 and A3. CBMs are most commonly associated with GHs, which have also been found in several PLs and GTs [14]. Only one CBM gene (CBM1) was upregulated in A2. GTs are enzymes that catalyze the formation of glycosidic linkages to form glycosides, which are involved in the biosynthesis of oligosaccharides, polysaccharides and glycoconjugates [35]. The GT8 gene was the most highly expressed in A3. PLs, also known as eliminases, are enzymes (EC 4.2.2.-) that cleave uronic acid-containing polysaccharides through a β-elimination mechanism, rather than via hydrolysis, to produce unsaturated polysaccharides [36]. In this study, the PL genes had relatively similar expression levels in the three treatments, indicating that PL expression might not be regulated by nitrogen levels. CEs represent a class of esterases that generally catalyze O-de- or N-deacylation to remove esters of substituted saccharides [37]. In this study, CEs had the highest expression levels in A1, and CE10 genes were predominant, with a total of 6 genes. The CE10 family has been found to act on noncarbohydrate substrates [38], suggesting that the latter are also upregulated by increased nitrogen levels in *S. rugosoannulata*. AA family proteins are mainly involved in the depolymerization of lignin or are found as primary cell wall contents in plants [27]. At the three nitrogen levels, the AA genes had different expression levels, suggesting that lignin degradation might not be affected by nitrogen levels such as cellulose and hemicellulose in *S. rugosoannulata*.

In the KEGG enrichment analysis, the MAPK signaling pathway was significantly enriched. MAPK signaling pathways are ubiquitous in eukaryotes and are involved in cellular development, differentiation, stress responses and nutrient utilization, among other processes [39]. In this study, only pheromone B α or β receptor genes were upregulated in A1. In basidiomycetes, pheromone B receptor genes regulate reciprocal nuclear exchange and nuclear migration in both sexes [40]. These results suggested that cell division was active in the A1 samples. In addition, the gene SHO1 was upregulated in A3; this gene is known as Hog1 and plays important roles in osmotic and oxidative stress responses [41]. In A1 and A2, SHO1 had similar expression levels, but its expression was significantly upregulated in A3, suggesting that increased nitrogen levels might cause more stress in *S. rugosoannulata*.

## 4. Materials and Methods

### 4.1. Collection of Stropharia rugosoannulata Materials at Different Nitrogen Levels

The *S. rugosoannulata* strain “DQ-1” (CGMCC5.2211) has been deposited in the China General Microbiological Culture Collection Center. The main component of the solid medium was corncob. The first solid medium (A1) consisted of 99% corncob and 1% calcium carbonate; the second solid medium (A2) consisted of 93% corncob, 6% soybean and 1% calcium carbonate; and the third solid medium (A3) consisted of 87% corncob, 12% soybean and 1% calcium carbonate. After complete mixing, the substrate was packed into polypropylene cultivation bags (average of 800 g/bottle with a moisture content of 65%), sterilized at 121 °C for 3 h and inoculated with a pure culture of *S. rugosoannulata*. Then, the cultivation bags were kept at 25 °C and 65–67% relative humidity (RH) in the dark for 30 days. Samples were collected from the three nitrogen levels and frozen at −80 °C; each sample had three biological replicates for RNA sequencing (RNA-seq) analysis and subsequent experiments.

### 4.2. Quantification of the Distance between Hyphal Branches

In addition, the three solid media (A1, A2 and A3) were also packed into glass culture dishes (average of 200 g/dish with a moisture content of 65%), sterilized at 121 °C for 3 h, and inoculated with a pure culture of *S. rugosoannulata*. The glass culture dishes were kept at 25 °C in the dark for two weeks, and images of the colony morphology were recorded using a Nikon camera Z30. The hyphae on the three solid media were viewed microscopically with a Zeiss LSM880 (Shanghai, Oberkochen) as described by Mu et al. [42]. The length between each pair of successive branches of leading hyphae was determined according to Ziv [43].

### 4.3. Total RNA Isolation, cDNA Library Preparation and Illumina Sequencing

Total RNA was extracted from the samples using TRIzol Reagent according to the manufacturer’s instructions (Invitrogen, Shanghai, China), and genomic DNA was removed using DNase I (TaKaRa, Dalian, China). The RNA band was clear; the 28/23S band was brighter than the 18/16S band; the RNA concentration was more than 20 ng/μL and the total RNA amount was more than 2 μg. After the RNA quality of these samples reached the standard, they were used to construct a sequencing library. RNA-seq transcriptome libraries were prepared using a TruSeqTM RNA Sample Preparation Kit from Illumina (San Diego, CA, USA), and the total RNA amount was 1 μg. PolyA selection with oligo(dT) beads and a fragmentation buffer were used to isolate and fragment the messenger RNA. Illumina-indexed adaptors were added according to Illumina’s protocol, which included cDNA synthesis, end repair, A-base addition and ligation. The cDNA target fragments of 200–300 bp were selected using 2% Low Range Ultra Agarose and subjected to PCR amplification using Phusion DNA polymerase (NEB) for 15 PCR cycles. The paired-end libraries were sequenced on an Illumina NovaSeq 6000 sequencing platform (150 bp*2, Shanghai BIOZERON Co., Ltd., Shanghai, China) after quantification using a TBS380. The datasets presented in this study can be found online in the NCBI repository under accession number PRJNA954980.

### 4.4. Read Quality Control and Mapping

First, the raw paired-end reads were trimmed and quality-controlled via Trimmomatic with the parameters SLIDINGWINDOW: 4:15 and MINLEN: 75 (version 0.36) (http://www.usadellab.org/cms/uploads/supplementary/trimmomatic, accessed on 22 June 2022). Then, the clean reads were separately aligned to the reference genome of *S. rugosoannulata* with orientation mode using HISAT2 software (https://ccb.jhu.edu/software/hisat2/index.shtml, accessed on 22 June 2022), and this software was also used to map default parameters. Later, qualimap_v2.2.1 (http://qualimap.bioinfo.cipf.es/, accessed on 22 June 2022) was used to assess the quality of these data. Finally, gene reads were counted using HTSeq (https://htseq.readthedocs.io/en/release_0.11.1/, accessed on 22 June 2022).

### 4.5. Differential Expression and Functional Enrichment Analyses

To identify differentially expressed genes (DEGs) between two different nitrogen levels in *S. rugosoannulata*, the fragments per kilobase of exon per million mapped reads (FPKM) method was used to calculate the expression level for each gene. The differential expression of each gene was analyzed using the R statistical package Empirical analysis of Digital Gene Expression in R (edgeR) (http://www.bioconductor.org/packages/release/bioc/html/edgeR.html/, accessed on 22 June 2022). The DEGs between two samples (A1 vs. A2, A2 vs. A3, A1 vs. A3) were selected using the following criteria: log (fold change) ≥ 2 and false discovery rate (FDR) < 0.05. Gene Ontology (GO) functional enrichment and Kyoto Encyclopedia of Genes and Genomes (KEGG) pathway enrichment analyses were carried out with GOATOOLS (https://github.com/tanghaibao/Goatools, accessed on 22 June 2022) and KOBAS (http://kobas.cbi.pku.edu.cn/home.do, accessed on 22 June 2022).

### 4.6. Determination of Cellulase and Hemicellulase Enzymes Activity

To detect the activity of cellulase and hemicellulase enzyems, the substrates at three different nitrogen levels were collected after *S. rugosoannulata* was cultured for 30 days in the mycelial vegetative stage. Then, 25 g samples of these substrates in the medium used for mycelium culture under the three nitrogen levels were added to 50 mL of distilled water and vibrated for 30 min at 200 rpm at 25 °C. The mixture was filtered using nonwoven fabric. The filtrate was centrifuged at 6000× *g* for 20 min at 4 °C, and the liquid supernatant was used to determine the activity of cellulose and hemicellulose enzymes. The activity and protein levels of these enzymes, namely, β-1,3-glucanase (β-GA), β-glucosaccharase (β-GC), β-1,4-glucanase (Cx) and xylanase (ACX), were determined using commercial assay kits (Keming Biotechnology Co., Ltd., Suzhou, China).

### 4.7. Determination of Nitrogen Metabolism Enzyme Activity

To detect the activity of the nitrogen-degrading enzymes, the substrates at three different nitrogen levels were collected after *S. rugosoannulata* was cultured for 30 days at the mycelial vegetative stage. Then, the samples were ground using liquid nitrogen. The activity and protein levels of nitrogen metabolism enzymes, namely, GOGAT, AS, GS, GDH, NiR and NR, were determined using commercial assay kits (Keming Biotechnology Co., Ltd., Suzhou, China).

### 4.8. Gene Expression Levels as Determined via qRT–PCR

To detect the gene expression levels, total RNA was isolated from samples using a TRIzol reagent (Takara, Dalian, China) according to the manufacturer’s instructions. Approximately 2 μg of total RNA from *S. rugosoannulata* at the three nitrogen levels (A1, A2 and A3) was reverse-transcribed via M-MLV reverse transcriptase (Takara) using oligo(dT) as described. qRT–PCR was performed as described by Hao et al. [44] using SYBR (Takara, Dalian, China). The primers and internal reference gene (18S ribosomal RNA) are listed in Appendix A. In addition, the relative gene expression was analyzed using the 2-ΔΔct method described by Livak and Schmittgen [45], and each experiment was performed in triplicate.

### 4.9. Statistical Analysis

The statistical significance analysis was completed using SPSS software (version 2.0) with Duncan’s multiple range test at a probability of *p* < 0.05.

## 5. Conclusions

Nitrogen sources are powerful nutrition factors regulating growth and lignocellulolytic enzyme production in fungi. In this study, three nitrogen levels (A1, A2 and A3) were designed for culturing *S. rugosoannulata*, and the results indicated that the mycelia were branched more highly and elongated more quickly in A3 than in A1 and A2. Most of the nitrogen metabolism-related enzymes had the highest activity in A1, but the cellulases had the highest activity in A3. Furthermore, transcriptome analysis of the DEGs showed that genes involved in the nitrogen metabolism pathway were upregulated in A1, while most of the genes involved in the starch and sucrose metabolism pathway, CAZyme pathway and MAPK signaling pathway were upregulated in A3. These results suggested that nitrogen metabolism and carbon metabolism have different responses to different nitrogen levels. Specifically, nitrogen metabolism in *S. rugosoannulata* is inhibited by elevated nitrogen, while carbon metabolism, especially cellulose metabolism, is induced by elevated nitrogen. These findings provide an understanding of the regulatory mechanism of nitrogen in cellulase production with straw-state substrate in basidiomycetes, which would help in establishing optimal culture strategies for better straw degradation in agriculture.

## Figures and Tables

**Figure 1 ijms-24-10089-f001:**
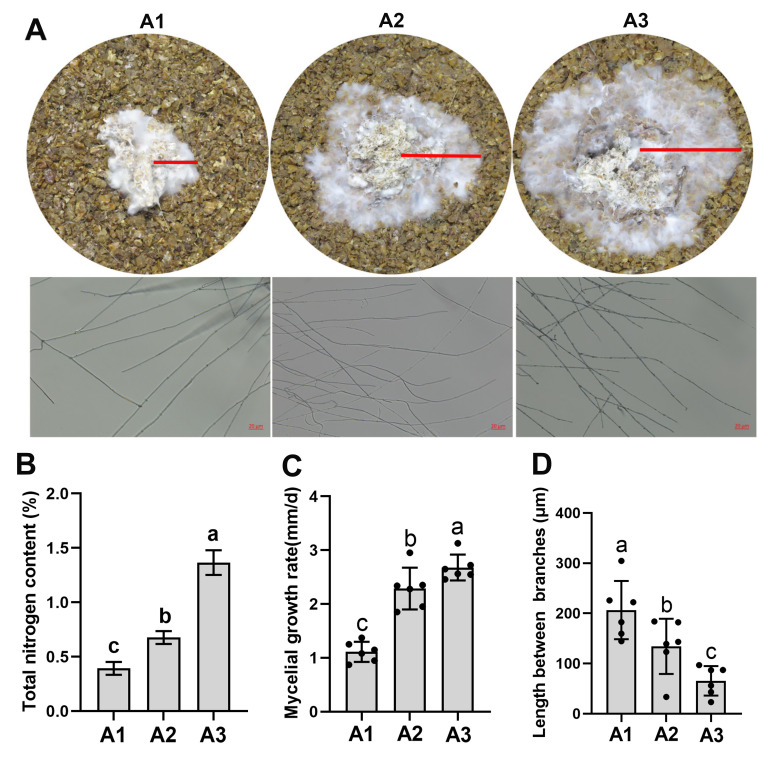
Different mycelial growth of *S. rugosoannulata* under three nitrogen levels. (**A**) Morphological features of mycelia on a solid medium under three nitrogen levels and mycelial branching in the tested solid media cultured at 25 °C (scale bar = 20 μm). (**B**) The three nitrogen levels of the solid media: A1: nitrogen content 0.39%; A2: nitrogen content 0.68%; A3: nitrogen content 1.37%. (**C**) The mycelial growth rate on the three solid media. (**D**) The distances between hyphal branches in the tested solid media. All data are presented as the means ± SDs from three independent experiments. Bars with different letters are significantly different at *p* < 0.05 according to Duncan’s multiple range test.

**Figure 2 ijms-24-10089-f002:**
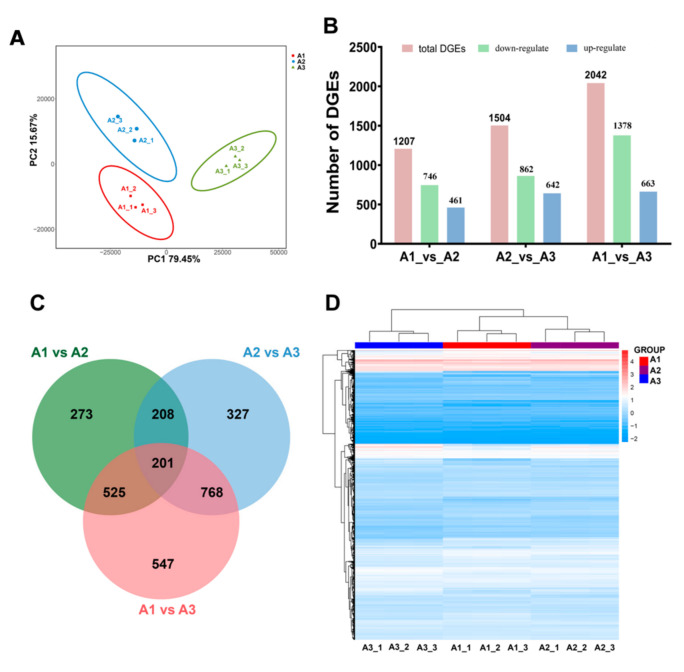
Analysis of DEGs under three nitrogen levels. (**A**) The PCA cluster diagram under the three nitrogen levels. (**B**) DEG distribution between the two samples analyzed. The number of DEGs is indicated at the top of the bar. (**C**) Venn diagrams comparing shared DEGs between the different nitrogen levels. (**D**) Expression patterns of 707 genes that overlapped among A1 vs. A2, A2 vs. A3 and A3 vs. A1.

**Figure 3 ijms-24-10089-f003:**
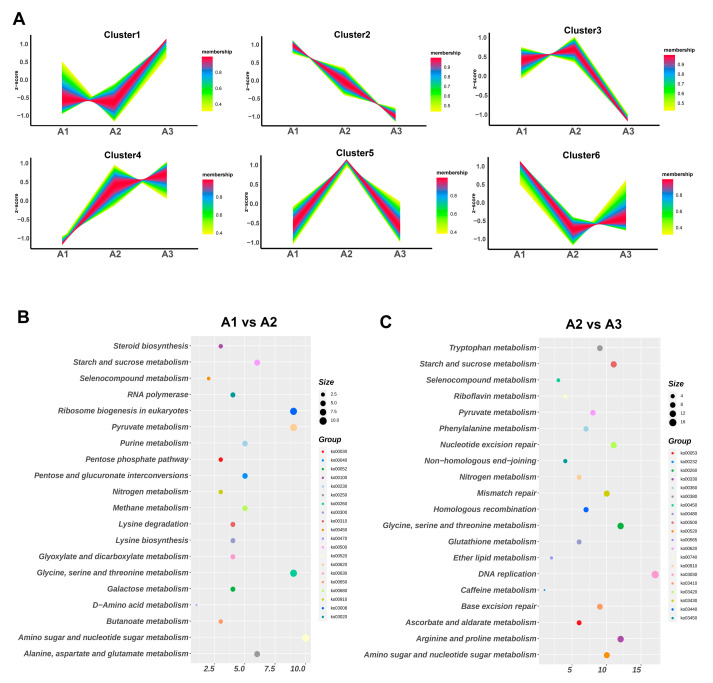
Expression clusters and KEGG enrichment of the DEGs. (**A**) The DEGs were divided into six different clusters. The top 20 KEGG pathways with the smallest FDR values, namely, the most significant enrichment among A1 vs. A2 (**B**) and A2 vs. A3 (**C**), were selected for display. The degree of enrichment is indicated by the rich factor, FDR value and number of genes enriched in the pathway. A larger rich factor indicates a greater degree of enrichment.

**Figure 4 ijms-24-10089-f004:**
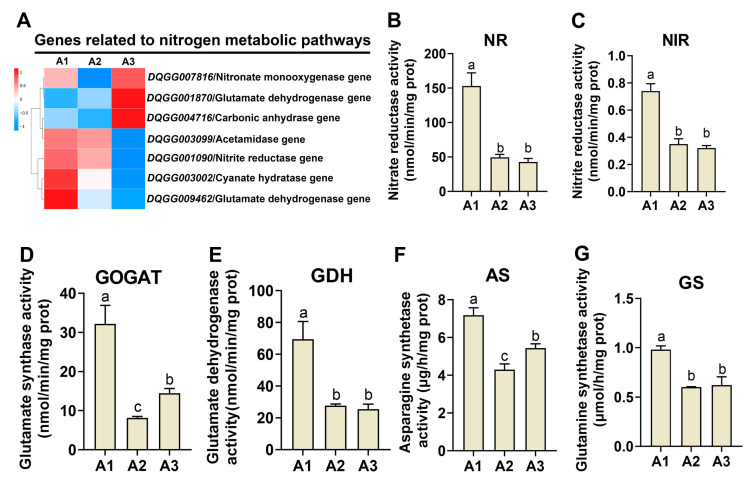
Gene expression pattern and activity analysis of nitrogen metabolism in *S. rugosoannulata*. (**A**) Gene expression profiles of the nitrogen metabolic pathway under three nitrogen levels. The enzyme activities of NR (**B**), NiR (**C**), GOGAT (**D**), GDH (**E**), AS (**F**) and GS (**G**) under three nitrogen levels. All data are presented as the means ± SDs of three independent experiments. Bars with different letters are significantly different at *p* < 0.05 according to Duncan’s multiple range test.

**Figure 5 ijms-24-10089-f005:**
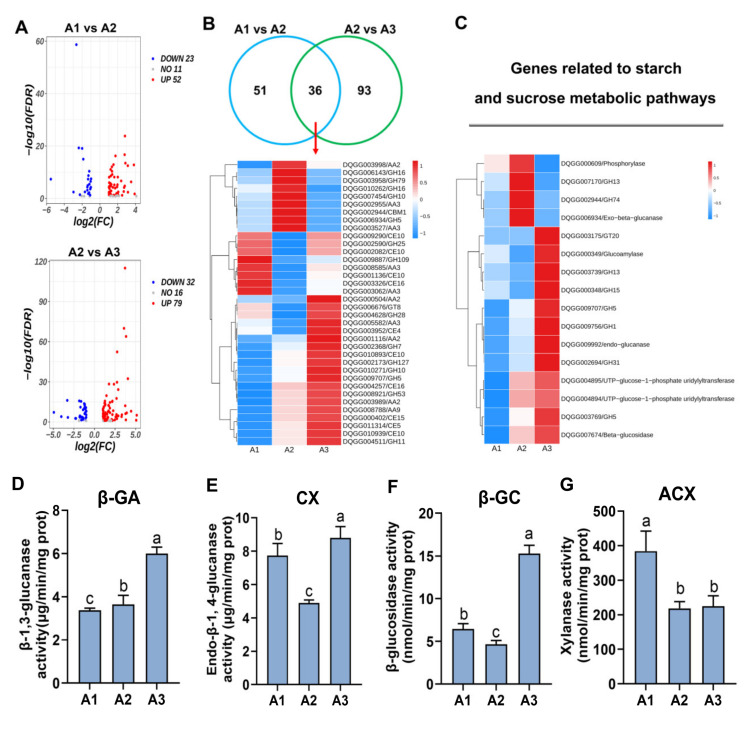
DEG and enzyme activity analysis of carbon metabolism at different nitrogen levels. (**A**) DEG distribution of the CAZyme genes at A1 vs. A2 and A2 vs. A3. UP represents the upregulated genes, DOWN represents the downregulated genes and NO represents the genes with no difference in expression. (**B**) Venn diagrams comparing shared DEGs between A1 vs. A2 and A2 vs. A3. A heatmap of the 36 overlapping genes in the Venn diagram was generated. (**C**) The gene expression profiles of the starch and sucrose metabolic pathways in A1, A2 and A3. (**D**–**G**) The activities of β-GA, Cx, β-GC and ACX were detected under three nitrogen levels. All data are presented as the means ± SDs of three independent experiments. Bars with different letters are significantly different at *p* < 0.05 according to Duncan’s multiple range test.

**Figure 6 ijms-24-10089-f006:**
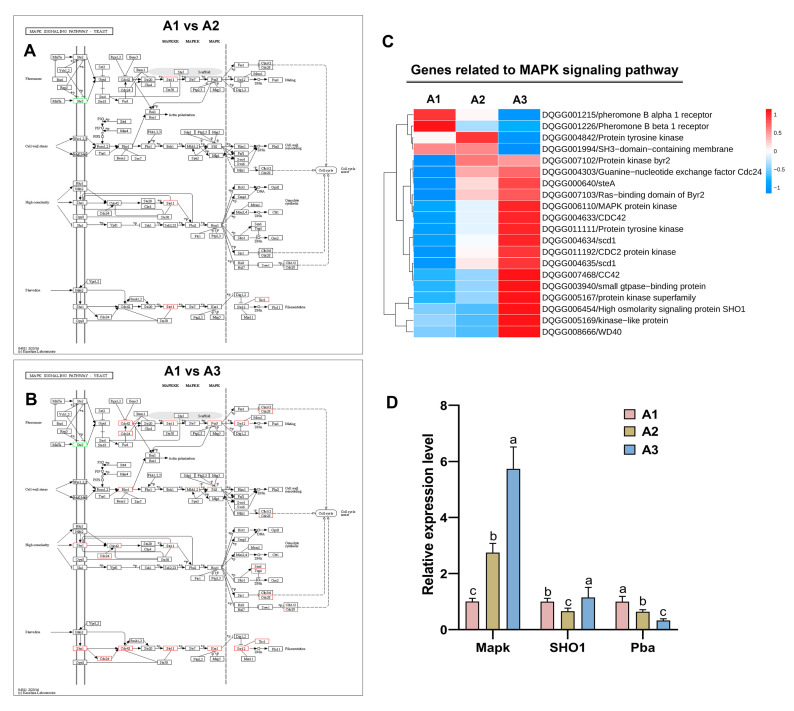
Putative pathway of MAPK signaling–yeast in *S. rugosoannulata* generated via KEGG analysis. (**A**) Gene expression profiles between the A1 and A2 growth stages. (**B**) Gene expression profiles between the A2 and A3 growth stages. The boxes with a red border indicate upregulation; the boxes with a green border represent downregulation. (**C**) Differential expression of MAPK signaling pathway–yeast genes at different nitrogen levels. (**D**) Validation of the gene expression levels of the MAPK signaling pathway. All data are presented as the means ± SDs of three independent experiments. Bars with different letters are significantly different at *p* < 0.05 according to Duncan’s multiple range test.

**Figure 7 ijms-24-10089-f007:**
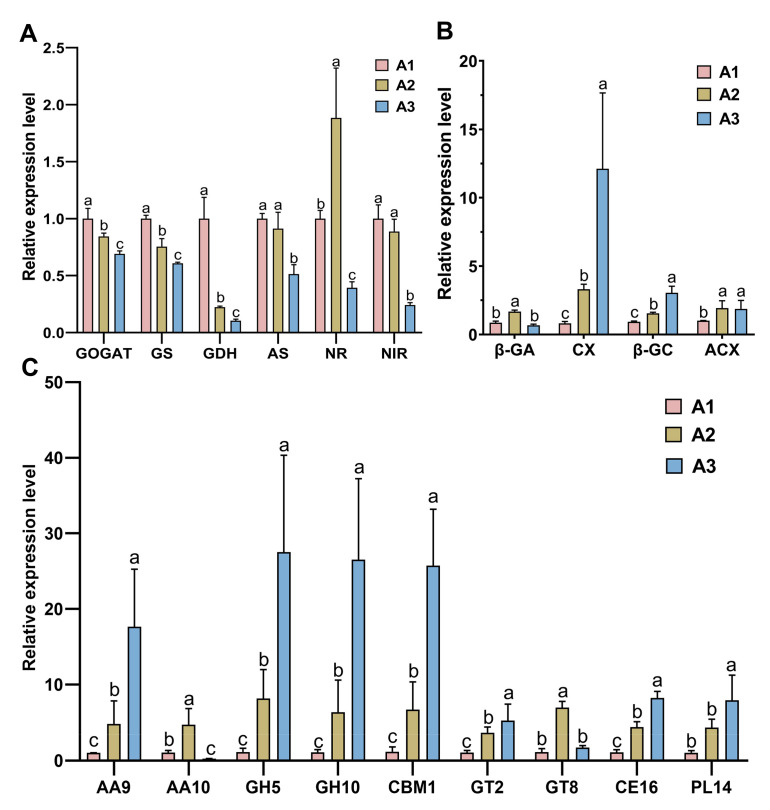
Validation of the gene expression levels of enzyme genes. (**A**) Six differentially expressed nitrogen metabolism genes (GOGAT, GS, GDH, AS, NR, and NiR) at the three nitrogen levels. (**B**) Five differentially expressed carbon metabolism genes (β-GA, β-GC, Cx and ACX) at the three nitrogen levels. (**C**) Ten differentially expressed CAZyme genes (AA10, AA9, GH5, GH10, CBM1, CNM50, GT2, GT8, CE16 and PL14) at the three nitrogen levels. All data are presented as the means ± SDs of three independent experiments. Bars with different letters are significantly different at *p* < 0.05 according to Duncan’s multiple range test.

**Table 1 ijms-24-10089-t001:** DEGs involved in the glycine, serine and threonine metabolism pathways in *S. rugosoannulata*.

GeneID	baseMean-A1	baseMean-A2	baseMean-A3	Pval	Gene Function
DQGG009599	533.06	550.04	145.56	8.44 × 10^−26^	glyoxylate reductase
DQGG008189	249.17	280.02	113.21	1.74 × 10^−6^	amine oxidase
DQGG003527	443.70	1328.30	440.56	3.0 × 10^−11^	glucose methanol choline oxidoreductase
DQGG005387	53.63	36.96	236.78	5.95 × 10^−27^	GMC oxidoreductase
DQGG003665	2045.36	2586.69	29347.66	2.8 × 10^−119^	GMC oxidoreductase
DQGG008582	1528.91	1674.92	20710.85	1.33 × 10^−67^	GMC oxidoreductase
DQGG001973	9854.39	7970.89	4091.93	1.96 × 10^−7^	amine oxidase
DQGG008585	338.21	99.35	248.68	9.11 × 10^−8^	GMC oxidoreductase
DQGG010007	2985.59	744.15	9090.15	1.07 × 10^−84^	cystathionine gamma lyase
DQGG003537	3070.79	1684.82	3702.9	0.0169	GMC oxidoreductase
DQGG005389	274.17	116.56	342.15	6.49 × 10^−7^	GMC oxidoreductase
DQGG005390	117.85	52.48	156.73	1.85 × 10^−5^	GMC oxidoreductase
DQGG008219	159.11	27.57	177.07	2.2 × 10^−6^	GMC oxidoreductase
DQGG000370	3016.03	2606.26	1380.65	2.25 × 10^−6^	amine oxidase
DQGG006088	2300.694	1127.07	816.63	0.1015	phosphoglycerate mutase
DQGG011096	5562.959	7671.32	13109.48	1.97 × 10^−11^	aryl-alcohol oxidase
DQGG001126	12,550.27	21,874.07	30,393.51	4.65 × 10^−6^	5-aminolevulinate synthase
DQGG004458	0.50	1.49	6.17	0.1174	pyranose dehydrogenase
DQGG008578	423.43	1536.30	1623.87	0.2570	GMC oxidoreductase
DQGG004502	437.49	1165.83	1718.57	0.0805	pyranose dehydrogenase
DQGG010568	463.16	1096.68	626.52	0.0001	GMC oxidoreductase
DQGG005392	1120.01	528.09	766.45	0.0065	GMC oxidoreductase
DQGG005391	312.05	153.79	179.9	0.0662	GMC oxidoreductase

baseMean (A1, A2 and A3) represents the read count homogenization results of this gene in the samples (A1, A2 and A3) of this group. Pval represents the significant *p* value.

## Data Availability

All data needed to support the conclusions are presented in this paper. Additional data related to this study were obtained from the authors.

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
