# Peer review of "Effects of Different Nitrogen Levels on Lignocellulolytic Enzyme Production and Gene Expression under Straw-State Cultivation in Stropharia rugosoannulata"

_ijms, 2023, doi:10.3390/ijms241210089_

Round 1

Reviewer 1 Report

The authors have evaluated the  Effects of three different nitrogen levels on the lignocellulolytic enzyme production and genes expression under straw-state cultivation in Stropharia rugosoannulata

 There are no line numbers in the manuscript that makes it difficult to point out errors

2.1. Include how different nitrogen levels were maintained and how old was the culture that was used for further experimentation? Samples were collected from the three nitrogen levels and frozen at -80°C. State those three levels.

subheading 2.2,  line 5 :  Nikon camera..add specifications please

subheading 2.4 , font size of heading  and weblinks  needs to be adjusted

subheading 2.9 <0.05 was regarded as statistically.?

subheading 3.3 line 4… rugusoannulate……….rugosoannulata

heading 4…abbrevaitions must not be used in headings and subheadings

heading 4 …second para line 4 Pleurotus dryinus…author citation is missing  for Pleurotus dryinus and  many other scientific  names.

Author Response

Dear Editor Aisa Safaya and Reviewers:

Thanks very much for your kind suggestion for my manuscript entitled “Effects of different nitrogen levels on the lignocellulolytic enzyme production and genes expression under straw-state cultivation in Stropharia rugosoannulata”(IJMS-2421182). We have carefully revised our manuscript according to your suggestions. The revised segment has been highlighted by red color in the revised manuscript. The point-to-point replies to these comments are listed below. We hope the revised manuscript will be suitable for publication.

Reviewer 1

The authors have evaluated the effects of three different nitrogen levels on the lignocellulolytic enzyme production and genes expression under straw-state cultivation in Stropharia rugosoannulata

There are no line numbers in the manuscript that makes it difficult to point out errors

Response: Thank you for your suggestion. We have added the line number in the revised manuscript.

2.1. Include how different nitrogen levels were maintained and how old was the culture that was used for further experimentation? Samples were collected from the three nitrogen levels and frozen at -80°C. State those three levels.

Response: We appreciated with these suggestions by the reviewers. In order to maintain the different nitrogen levels of the solid medium, we firstly choose the same supplier for supplying corncob and soybean which the manufacturing process is relative stable, and then the three nitrogen levels of the solid media were made as the same ratio, and the solid media were stirred fully to make the solid media well-distributed. S. rugosoannulata was cultured for 30 days for further experimentation (Page 6, Lines 122-124). The samples which included the mycelia and solid medium were collected while the mycelia were cultured for 30 days and then were frozen in liquid nitrogen for at least 5 min and were kept at -80℃ for further experimentation.

subheading 2.2, line 5:  Nikon camera..add specifications please

Response: Thank you very much. We have added the specification in the revised manuscript (Page 6, Line 131).

subheading 2.4, font size of heading and weblinks needs to be adjusted

Response: Thank you very much! We have revised the font size of the heading and weblinks in the revised manuscript (Page 7, Lines 154-157).

subheading 2.9 <0.05 was regarded as statistically.?

Response: I am sorry for the inaccurate expression. We have revised the writing in the revised manuscript (Page 10, Lines 207-208).

subheading 3.3 line 4… rugusoannulate……….rugosoannulata

Response: Thank you very much! We have revised the writing in the revised manuscript (Page 12, Line 259).

heading 4…abbrevaitions must not be used in headings and subheadings

Response: We appreciated with these suggestions. We have written full names in the headings and subheadings in the revised manuscript (Page 11. Line 225; Page 12, Lines 254-255; Page 18, Line 396).

heading 4 …second para line 4 Pleurotus dryinus…author citation is missing for Pleurotus dryinus and many other scientific names.

Response: We appreciated with this suggestion. We have added the scientific names of fungi in the revised manuscript (Page 21, Line 443-445).

Thanks very much for your attention to our manuscript!

Best wishes!

Hui Chen

6, Jun, 2023

Reviewer 2 Report

Comments to the Author:

Title: Effects of different nitrogen levels on the lignocellulolytic enzyme production and genes expression under straw-state cultivation in Stropharia rugosoannulata

Overview and general recommendation:

The manuscript deals with an important topic related to the effects of different nitrogen levels on the lignocellulolytic enzyme production and genes expression under straw-state cultivation in Stropharia rugosoannulata. The manuscript technically sounds well and shows high novelty. However, it needs major linguistic adjustments; therefore, I invite the authors to pass their manuscript to a native English speaker for editing and revision. In this regard, the needed adjustments are highlighted in “Minor comments” section. Line numbering should be added as the review was tough without it. The references used to support statements shall be updated to be from 2010 and onwards so as to be more reliable in the current times.

The Abstract part outlines clearly the problematic, aims, methodology and findings of the current study while reporting the main conclusions aroused. The Introduction part is well structured and aiming and underlines appropriately the whole subject under study. The aims of the study are also clear and understood. The Materials and methods part is clear, well written, and encloses most of the information related to the adopted methodology (minor clarifications are needed), and statistical analysis. Although it shows a correct statistical representation, the Results part needs major adjustments. The scientific analysis of the findings shall be improved. The Discussion part is well appropriate and compares the current findings with previous studies in literature. However, the duplication of findings shall be dealt with and avoided. An appropriate Conclusions part was added in which authors summarized the findings of their study and suggested further related research being based on the raised assumptions.

My comments and queries for authors are detailed below in “Major comments” and “Minor comments” sections.

1.1.            Major comments:

1-      The manuscript needs major linguistic adjustments; accordingly, I invite the authors to pass their manuscript to a native English speaker for editing and revision. Most needed adjustments are highlighted in “Minor comments” section.

2-      The references used to support statements shall be updated to be from 2010 and onwards so as to be more reliable in the current times.

3-      Results: The scientific analysis of the findings shall be improved.

4-      Discussion: The duplication of findings outline in this part shall be avoided.

1.2.            Minor comments:

5-      Abstract, Page 1: Kindly adjust the sentence as follow: “The nitrogen and carbon metabolisms are the most…”

6-      Abstract, Page 1: Kindly remove “In this study”.

7-      Abstract, Page 1: Kindly adjust as follow: “The activity of nitrogen… was higher”.

8-      Abstract, Page 1: “The activity of nitrogen… higher in A1”: The sentence is cumbersome; accordingly, kindly reformulate in order to make it clearer and more aiming.

9-      Abstract, Page 1: Kindly adjust as follow: “starch and sucrose metabolisms”.

10-  Abstract, Page 1: Kindly adjust as follow: “can upregulate”.

11-  Abstract, Page 1: Kindly adjust as follow: “Basidiomycetes”.

12-  1. Introduction, Page 1: Kindly adjust as follow: “cultivated on straw”.

13-  1. Introduction, Page 1: Kindly adjust the sentence as follow: “The main carbon source… is lignocellulose… which consists of…”

14-  1. Introduction, Page 1: Kindly adjust as follow: “Nitrogen is another”.

15-  1. Introduction, Page 1: Kindly adjust as follow: “lignocellulosic degradation”.

16-  1. Introduction, Page 2: Kindly adjust as follow: “is still unclear”.

17-  1. Introduction, Page 2: Kindly adjust as follow: “fungi can uptake”.

18-  1. Introduction, Page 2: Kindly adjust as follow: “which involve multiple”.

19-  1. Introduction, Page 2: “Asparagine synthetase… involved”: How are they involved?

20-  1. Introduction, Page 2: “In yeast… [8]”: How is it important?

21-  1. Introduction, Page 2: “Carbon is another… [9]”: The reference used for this statement is relatively very old (older than 2010); accordingly, kindly replace it by the following reliable and recent one: “doi:10.1088/1755-1315/1090/1/012020”.

22-  1. Introduction, Page 2: “Both excess… [13-14]”: The references used for this statement are relatively very old (older than 2010); accordingly, kindly replace them by the following reliable and recent one: “doi:10.3390/agriculture12122095”.

23-  1. Introduction, Page 2: “Nitrogen sources are usually… xylanases [17-19]”: References “[15]”, “[16]”, “[18]”, and “[19]” are relatively very old (older than 2010); accordingly, kindly replace them by more recent studies that are plenty in literature.

24-  1. Introduction, Page 2: Same recommendation as the previous comment regarding reference “[21]”.

25-  1. Introduction, Page 2: “Overall… biodegradation efficiency”: The sentence is cumbersome; accordingly, kindly reformulate in order to make it clearer and more aiming.

26-  2. Materials and Methods, 2.2. Quantification of the distance between hyphal branches, Page 2: Kindly mention the full specification of the microscope (company, city and country of origin).

27-  2. Materials and Methods, 2.4. Read quality control and mapping, Page 3: “Then… default parameters”: The sentence is cumbersome; accordingly, kindly reformulate in order to make it clearer and more aiming.

28-  2. Materials and Methods, 2.4. Read quality control and mapping, Page 3: Kindly remove “each” after “Finally”.

29-  2. Materials and Methods, 2.5. Differential expression and functional enrichment analyses, Page 3: “To identify… expression levels”: The sentence is cumbersome; accordingly, kindly reformulate in order to make it clearer and more aiming.

30-  2. Materials and Methods, 2.6. Determination of cellulase and hemicellulase activity, Page 4: Kindly adjust as follow: “activity of cellulase and hemicellulase enzymes”.

31-  2. Materials and Methods, 2.7. Determination of nitrogen metabolism enzyme activity, Page 4: Kindly adjust as follow: “To detect the activity”.

32-  2. Materials and Methods, 2.8. Gene expression levels as determined by qRT-PCR, Page 4: Kindly adjust as follow: “as described”.

33-  2. Materials and Methods, 2.9. Statistical analysis, Page 4: Kindly remove “All” and adjust as follow: “as statistically significant”.

34-  3. Results, 3.1. Nitrogen levels regulated the mycelial growth of S. rugosoannulata, Page 4: Kindly adjust as follow: “and A3 were noted after inoculation…”

35-  3. Results, 3.1. Nitrogen levels regulated the mycelial growth of S. rugosoannulata, Page 4: Kindly add “respectively” after “1.37%”.

36-  3. Results, 3.1. Nitrogen levels regulated the mycelial growth of S. rugosoannulata, Page 4: “The mycelial rate… and A2”: The sentence is badly written in standard English; accordingly, kindly reformulate it.

37-  3. Results, 3.2. Global transcriptomic analysis of S. rugosoannulata and identification of DEGs, Page 5: Kindly adjust as follow: “based on”.

38-  3. Results, 3.2. Global transcriptomic analysis of S. rugosoannulata and identification of DEGs, Page 5: “The contribution… A2 vs A3”: The sentence is cumbersome; accordingly, kindly reformulate in order to make it clearer and more aiming.

39-  3. Results, 3.3. KEGG pathway and GO enrichment analyses of the DEGs, Page 7: Kindly adjust as follow: “the genes” and “S. rugosoannulata”.

40-  3. Results, 3.3. KEGG pathway and GO enrichment analyses of the DEGs, Page 7: “The enriched… metabolism (Figure S1B)”: The sentence is badly written in standard English; accordingly, kindly reformulate it.

41-  3. Results, 3.3. KEGG pathway and GO enrichment analyses of the DEGs, Page 8: Kindly replace “elevated” by “increased”.

42-  3. Results, 3.4. Analysis of nitrogen metabolic genes at the three nitrogen levels, Page 9: “Therefore… S. rugosoannulata”: Kindly avoid the first voice form of the sentence and adopt the impersonal form instead.

43-  3. Results, 3.4. Analysis of nitrogen metabolic genes at the three nitrogen levels, Page 9: Kindly adjust as follow: “and only one gene… was upregulated”.

44-  3. Results, 3.4. Analysis of nitrogen metabolic genes at the three nitrogen levels, Page 9: Kindly adjust as follow: “in response to”.

45-  3. Results, 3.4. Analysis of nitrogen metabolic genes at the three nitrogen levels, Page 9: “We thus… nitrogen levels”: Kindly avoid the first voice form of the sentence and adopt the impersonal form instead.

46-  3. Results, 3.4. Analysis of nitrogen metabolic genes at the three nitrogen levels, Page 9: Kindly adjust as follow: “no significant differences”.

47-  3. Results, 3.4. Analysis of nitrogen metabolic genes at the three nitrogen levels, Page 9: Kindly adjust as follow: “than the other two samples”.

48-  3. Results, 3.4. Analysis of nitrogen metabolic genes at the three nitrogen levels, Page 9: “Besides… than A2”: The sentence is badly written in standard English; accordingly, kindly reformulate it.

49-  3. Results, 3.4. Analysis of nitrogen metabolic genes at the three nitrogen levels, Page 9: “The activity of GDH… metabolism enzymes”: Same recommendation as in the previous comments.

50-  3. Results, 3.4. Analysis of nitrogen metabolic genes at the three nitrogen levels, Page 10: Kindly remove “significantly” before “differentially”.

51-  3. Results, 3.5. Analysis of carbon metabolic genes at the three nitrogen levels, Page 11: Kindy adjust the numbering of this paragraph as “3.5.” instead of “3.4.”

52-  3. Results, 3.5. Analysis of carbon metabolic genes at the three nitrogen levels, Page 11: Kindly remove reference “[5]” from here.

53-  3. Results, 3.5. Analysis of carbon metabolic genes at the three nitrogen levels, Page 11: “In this study… (Figure 5A)”: Kindly avoid the first voice form of the sentence and adopt the impersonal form instead.

54-  3. Results, 3.5. Analysis of carbon metabolic genes at the three nitrogen levels, Page 11: Kindly adjust as follow: “no significant differences”.

55-  3. Results, 3.5. Analysis of carbon metabolic genes at the three nitrogen levels, Page 11: “Furthermore… (Figure 5B)”: Kindly avoid the first voice form of the sentence and adopt the impersonal form instead.

56-  3. Results, 3.5. Analysis of carbon metabolic genes at the three nitrogen levels, Page 11: “During these genes… in A1”: The sentence is badly written in standard English; accordingly, kindly reformulate it.

57-  3. Results, 3.5. Analysis of carbon metabolic genes at the three nitrogen levels, Page 12: “We also analyzed… (Figure 5B)”: Kindly avoid the first voice form of the sentence and adopt the impersonal form instead.

58-  3. Results, 3.5. Analysis of carbon metabolic genes at the three nitrogen levels, Page 12: Kindly adjust as follow: “were upregulated”.

59-  3. Results, 3.5. Analysis of carbon metabolic genes at the three nitrogen levels, Page 13: “Therefore… S. rugosoannulata”: Kindly avoid the first voice form of the sentence and adopt the impersonal form instead.

60-  3. Results, 3.5. Analysis of carbon metabolic genes at the three nitrogen levels, Page 13: Kindly adjust as follow: “were the highest in A2”.

61-  3. Results, 3.5. Analysis of carbon metabolic genes at the three nitrogen levels, Page 13: Kindly adjust as follow: “with the analyses of carbohydrate”.

62-  3. Results, 3.5. Analysis of carbon metabolic genes at the three nitrogen levels, Page 13: “We also detected… [18-19]”: Kindly move this statement to the Discussion part. Also, kindly avoid the first voice form of the sentence and adopt the impersonal form instead.

63-  3. Results, 3.5. Analysis of carbon metabolic genes at the three nitrogen levels, Page 13: “We found… (Figure 5G)”: Kindly avoid the first voice form of the sentence and adopt the impersonal form instead.

64-  3. Results, 3.6. Analysis of the MAPK signaling pathway at the three nitrogen levels, Page 13: Kindly adjust the numbering of this paragraph as “3.6.: instead of “3.5.”

65-  3. Results, 3.6. Analysis of the MAPK signaling pathway at the three nitrogen levels, Page 13: “We found that… (DQGG008666)”: Kindly avoid the first voice form of the sentence and adopt the impersonal form instead. Moreover, the sentence is long and cumbersome; accordingly, kindly reformulate in order to make it more concise, clearer and more aiming.

66-  3. Results, 3.6. Analysis of the MAPK signaling pathway at the three nitrogen levels, Page 13: “Besides… (Figure 6D)”: Kindly avoid the first voice form of the sentence and adopt the impersonal form instead.

67-  3. Results, 3.7. Validation of transcriptomic data by qRT-PCR, Page 14: Kindly adjust the numbering of this paragraph as “3.7.” instead of “3.6.”

68-  3. Results, 3.7. Validation of transcriptomic data by qRT-PCR, Page 14: Kindly adjust as follow: “similarly to”.

69-  3. Results, 3.7. Validation of transcriptomic data by qRT-PCR, Pages 14–15: “In addition… (Figure 7C)”: Kindly avoid the first voice form of the sentence and adopt the impersonal form instead.

70-  3. Results, 3.7. Validation of transcriptomic data by qRT-PCR, Page 15: Kindly adjust as follow: “similarly to”.

71-  4. Discussion, Page 16: “S. rugosoannulata… in fungi [4]”: Kindly remove these sentences as you already mentioned that in the Introduction part.

72-  4. Discussion, Page 16: “In this study… quickly in A3”: Kindly avoid the first voice form of the sentence and adopt the impersonal form instead.

73-  4. Discussion, Page 16: “In addition… inhibited in A3”: Same recommendation as in the previous comment.

74-  4. Discussion, Page 16: Kindly adjust as follow: “nitrogen regulates”.

75-  4. Discussion, Page 16: “In fungi… ACXs [15,18-19]”: It is better to replace these old references by newer ones as aforementioned.

76-  4. Discussion, Page 16: “In Pleurotus dryinus… [16]”: The sentence is badly written in standard English; accordingly, kindly reformulate it. Also, same recommendation as in the previous comment.

77-  4. Discussion, Page 16: “Kapich et al. [27] … ME-446”: Same point regarding the old references.

78-  4. Discussion, Page 16: Kindly adjust as follow: “On the contrary”.

79-  4. Discussion, Page 16: “From these results… effects in fungi”: Kindly avoid the first voice form of the sentence and adopt the impersonal form instead. Moreover, kindly reformulate the sentence in a better language.

80-  4. Discussion, Page 16: Kindly adjust as follow: “Similarly to the case of”.

81-  4. Discussion, Page 16: Kindly adjust as follow: “substrate of”.

82-  4. Discussion, Page 16: “Nitrogen metabolism… of nitrogen sources”: The sentence is badly written in standard English; accordingly, kindly reformulate it.

83-  4. Discussion, Page 17: “CEs represent… saccharides [37]”: Same point regarding the old references.

84-  4. Discussion, Page 17: “The CE10 family… levels in S. rugosoannulata”: Same recommendation as in the previous comment.

85-  4. Discussion, Page 17: “During the KEGG… significantly enriched”: Kindly avoid the first voice form of the sentence and adopt the impersonal form instead.

86-  5. Conclusions, Page 17: “In this study… than the A1 and A2”: Same recommendation as in the previous comment.

The manuscript needs major linguistic adjustments; therefore, I invite the authors to pass their manuscript to a native English speaker for editing and revision. In this regard, the needed adjustments are highlighted in “Minor comments” section in the attached report.

Author Response

Dear Editor Aisa Safaya and Reviewers:

Thanks very much for your kind suggestion for my manuscript entitled “Effects of different nitrogen levels on the lignocellulolytic enzyme production and genes expression under straw-state cultivation in Stropharia rugosoannulata”(IJMS-2421182). We have carefully revised our manuscript according to your suggestions. The revised segment has been highlighted by red color in the revised manuscript. The point-to-point replies to these comments are listed below. We hope the revised manuscript will be suitable for publication.

Reviewers2

Overview and general recommendation:

The manuscript deals with an important topic related to the effects of different nitrogen levels on the lignocellulolytic enzyme production and genes expression under straw-state cultivation in Stropharia rugosoannulata. The manuscript technically sounds well and shows high novelty. However, it needs major linguistic adjustments; therefore, I invite the authors to pass their manuscript to a native English speaker for editing and revision. In this regard, the needed adjustments are highlighted in “Minor comments” section. Line numbering should be added as the review was tough without it. The references used to support statements shall be updated to be from 2010 and onwards so as to be more reliable in the current times.

Response: We appreciate the positive comments about the manuscript. We consider these suggestions are all significant for our research work and paper writing. We have sent our manuscript to the AJE company to edit the language and it may be verified

on the AJE website using the verification code A810-F462-5BDB-2610-900P. Besides, the line numbering has been added in the revised manuscript. The references used to support statements have been updated to be from 2010 in the revised manuscript (Pages 27-32, Lines 574-575; 579-582; 594-608; 612-618; 622-624; 642-645; 653-661; 681-686).

The Abstract part outlines clearly the problematic, aims, methodology and findings of the current study while reporting the main conclusions aroused. The Introduction part is well structured and aiming and underlines appropriately the whole subject under study. The aims of the study are also clear and understood. The Materials and methods part is clear, well written, and encloses most of the information related to the adopted methodology (minor clarifications are needed), and statistical analysis. Although it shows a correct statistical representation, the Results part needs major adjustments. The scientific analysis of the findings shall be improved.

Response: We appreciate the positive comments about the manuscript. We have revised the results part in the revised manuscript (Pages 10-20; Lines 211-425).

The Discussion part is well appropriate and compares the current findings with previous studies in literature. However, the duplication of findings shall be dealt with and avoided. An appropriate Conclusions part was added in which authors summarized the findings of their study and suggested further related research being based on the raised assumptions.

Response: We appreciate the positive comments about the manuscript. We have revised the Discussion part and the duplication of findings have been dealt with and avoided in the revised manuscript (Pages 20-24; Lines 428-517).

My comments and queries for authors are detailed below in “Major comments” and “Minor comments” sections.

1.1.            Major comments:

  • The manuscript needs major linguistic adjustments; accordingly, I invite the authors to pass their manuscript to a native English speaker for editing and revision. Most needed adjustments are highlighted in “Minor comments” section.

Response: We appreciated with these suggestions. We have sent our manuscript to the AJE company to edit the language and it may be verified on the AJE website using the verification code A810-F462-5BDB-2610-900P.

  • The references used to support statements shall be updated to be from 2010 and onwards so as to be more reliable in the current times.

Response: We appreciated with this suggestion by the Reviewer. The references used to support statements have been updated to be from 2010 in the revised manuscript (Pages 27-32, Lines 574-575; 579-582; 594-608; 612-618; 622-624; 642-645; 653-661; 681-686).

  • Results: The scientific analysis of the findings shall be improved.

Response: We appreciated with this suggestion by the Reviewer. We have revised the results part in the revised manuscript (Pages 10-20; Lines 211-425).

  • Discussion: The duplication of findings outline in this part shall be avoided.

Response: We appreciated with this suggestion by the Reviewer. We have revised the Discussion part and the duplication of findings have been dealt with and avoided in the revised manuscript (Pages 20-24; Lines 428-517).

1.2.            Minor comments:

  • Abstract, Page 1: Kindly adjust the sentence as follow: “The nitrogen and carbon metabolisms are the most…”

Response: Thank you very much. We have revised it in the revised manuscript (Page 2, Line 24).

  • Abstract, Page 1: Kindly remove “In this study”.

Response: Thank you! We have removed “In this study” in the revised manuscript.

  • Abstract, Page 1: Kindly adjust as follow: “The activity of nitrogen… was higher”.

Response: Thank you very much! We have revised it in the revised manuscript (Page 2, Line 32).

  • Abstract, Page 1: “The activity of nitrogen… higher in A1”: The sentence is cumbersome; accordingly, kindly reformulate in order to make it clearer and more aiming.

Response: We appreciated with this suggestion. We have rewritten this sentence to make it clearer and more aiming in the revised manuscript (Page 2, Line 32-33).

  • Abstract, Page 1: Kindly adjust as follow: “starch and sucrose metabolisms”.

Response: Thank you! We have revised the writing as follow “starch and sucrose metabolisms” in the revised manuscript (Page 2, Line 29).

  • Abstract, Page 1: Kindly adjust as follow: “can upregulate”.

Response: Thank you very much! We have revised it as follow “can upregulate” in the revised manuscript (Page 2, Lines 37-38).

  • Abstract, Page 1: Kindly adjust as follow: “Basidiomycetes”.

Response: We appreciated with this suggestion. We have revised it as follow “Basidiomycetes” in the revised manuscript (Page 2, Line 40).

  • Introduction, Page 1: Kindly adjust as follow: “cultivated on straw”.

Response: Thank you! We have revised it as follow “cultivated on straw” in the revised manuscript (Page 3, Line 49).

  • Introduction, Page 1: Kindly adjust the sentence as follow: “The main carbon source… is lignocellulose… which consists of…”

Response: Thank you very much! We have revised this sentence in the revised manuscript (Page 3, Lines 55-56).

  • Introduction, Page 1: Kindly adjust as follow: “Nitrogen is another”.

Response: Thank you very much! We have revised it as follow “Nitrogen is another” in the revised manuscript (Page 3, Line 57).

  • Introduction, Page 1: Kindly adjust as follow: “lignocellulosic degradation”.

Response: Thank you! We have revised the writing as follow “lignocellulosic degradation” in the revised manuscript (Page 3, Line 58).

  • 1. Introduction, Page 2: Kindly adjust as follow: “is still unclear”.

Response: Thank you! We have revised it in the revised manuscript (Page 3, Line 59).

  • Introduction, Page 2: Kindly adjust as follow: “fungi can uptake”.

Response: Thank you! We have revised it as the suggestion in the revised manuscript (Page 3, Line 64).

  • Introduction, Page 2: Kindly adjust as follow: “which involve multiple”.

Response: Thank you very much! We have revised it in the revised manuscript (Page 3, Line 65).

  • Introduction, Page 2: “Asparagine synthetase… involved”: How are they involved?

Response: We appreciated with this suggestion. Asparagine synthetase catalyses the transfer of an amino group from glutamine to aspartate to form glutamate and asparagine, which is crucial for glutamine metabolism in Saccharomyces cerevisiae. We have added the descriptions in the revised manuscript (Page 4, Lines 71-74).

  • Introduction, Page 2: “In yeast… [8]”: How is it important?

Response: We appreciated with this suggestion. In yeast, the central nitrogen metabolism hosts the two conserved mechanism pathways for glutamate production. The GS-GOGAT pathway has a marginal contribution in glutamate synthesis both in fermentation and respiratory conditions, while the GDH pathway has the prominent role. In the GDH pathway, the synthesis of glutamate using α-ketoglutarate and ammonium through the NADP linked action of GDH. Therefore, the GDH pathway is important for nitrogen metabolism. We have added the descriptions in the revised manuscript (Page 4, Lines 74-77).

  • Introduction, Page 2: “Carbon is another… [9]”: The reference used for this statement is relatively very old (older than 2010); accordingly, kindly replace it by the following reliable and recent one: “doi:10.1088/1755-1315/1090/1/012020”.

Response: Thank you very much for providing the reference for us. However, we could not find the reference by using the “doi:10.1088/1755-1315/1090/1/012020”. Please provide me the Doi number or the title of the reference again! Besides, we also added another reference in the revised manuscript (Page 27, Lines 588-590).

  • Introduction, Page 2: “Both excess… [13-14]”: The references used for this statement are relatively very old (older than 2010); accordingly, kindly replace them by the following reliable and recent one: “doi:10.3390/agriculture12122095”.

Response: Thank you very much for this suggestion and for providing the reference. We have renewed the reference in the revised manuscript (Page 28, Lines 601-609).

  • Introduction, Page 2: “Nitrogen sources are usually… xylanases [17-19]”: References “[15]”, “[16]”, “[18]”, and “[19]” are relatively very old (older than 2010); accordingly, kindly replace them by more recent studies that are plenty in literature.

Response: We appreciated with these suggestions. We have renewed these references in the revised manuscript (Page 28, Lines 610-615; Page 29, lines 619-626).

  • Introduction, Page 2: Same recommendation as the previous comment regarding reference “[21]”.

Response: We appreciated with this suggestion. We have renewed the reference and the manuscript in the revised manuscript (Page 5, Lines 98-100; Page 29, Lines 630-632).

  • Introduction, Page 2: “Overall… biodegradation efficiency”: The sentence is cumbersome; accordingly, kindly reformulate in order to make it clearer and more aiming.

Response: We appreciated with this suggestion. We have revised the sentences in the revised manuscript (Page 5, Lines 107-110).

  • Materials and Methods, 2.2. Quantification of the distance between hyphal branches, Page 2: Kindly mention the full specification of the microscope (company, city and country of origin).

Response: Thank you very much! We have added the full specification of the microscope in the revised manuscript (Page 6, Lines 132).

  • Materials and Methods, 2.4. Read quality control and mapping, Page 3: “Then… default parameters”: The sentence is cumbersome; accordingly, kindly reformulate in order to make it clearer and more aiming.

Response: We appreciated with these suggestions. We have revised these sentences to make it clearer and more aiming in the revised manuscript (Page 7, Line 154; Page 8, Lines 157-161).

  • Materials and Methods, 2.4. Read quality control and mapping, Page 3: Kindly remove “each” after “Finally”.

Response: Thank you! We have removed “each” in the revised manuscript.

  • Materials and Methods, 2.5. Differential expression and functional enrichment analyses, Page 3: “To identify… expression levels”: The sentence is cumbersome; accordingly, kindly reformulate in order to make it clearer and more aiming.

Response: We appreciated with this suggestion. We have revised this sentence to make it clearer and more aiming in the revised manuscript (Page 8, Lines 165-167).

  • Materials and Methods, 2.6. Determination of cellulase and hemicellulase activity, Page 4: Kindly adjust as follow: “activity of cellulase and hemicellulase enzymes”.

Response: Thank you very much! We have revised it in the revised manuscript (Page 9, Line 178).

  • Materials and Methods, 2.7. Determination of nitrogen metabolism enzyme activity, Page 4: Kindly adjust as follow: “To detect the activity”.

Response: Thank you! We have revised it in the revised manuscript (Page 9, Line 190)

  • Materials and Methods, 2.8. Gene expression levels as determined by qRT-PCR, Page 4: Kindly adjust as follow: “as described”.

Response: Thank you! We have revised it in the revised manuscript (Page 10, Line 201).

  • Materials and Methods, 2.9. Statistical analysis, Page 4: Kindly remove “All” and adjust as follow: “as statistically significant”.

Response: Thank you very much! We have revised this part in the revised manuscript (Page 10, Lines 207-208).

  • Results, 3.1. Nitrogen levels regulated the mycelial growth of S. rugosoannulata, Page 4: Kindly adjust as follow: “and A3 were noted after inoculation…”

Response: Thank you very much! We have revised this sentence to make it clearer and more aiming in the revised manuscript (Page 10, Lines 217-218).

  • Results, 3.1. Nitrogen levels regulated the mycelial growth of S. rugosoannulata, Page 4: Kindly add “respectively” after “1.37%”.

Response: Thank you very much! We have revised it in the manuscript (Page 10, Line 217).

  • Results, 3.1. Nitrogen levels regulated the mycelial growth of S. rugosoannulata, Page 4: “The mycelial rate… and A2”: The sentence is badly written in standard English; accordingly, kindly reformulate it.

Response: Thank you very much! We have reformulated this sentence in the revised manuscript (Page 10, Lines 217-219).

  • Results, 3.2. Global transcriptomic analysis of S. rugosoannulataand identification of DEGs, Page 5: Kindly adjust as follow: “based on”.

Response: Thank you very much! We have revised this word in the revised manuscript (Page 11, Line 227).

  • Results, 3.2. Global transcriptomic analysis of S. rugosoannulataand identification of DEGs, Page 5: “The contribution… A2 vs A3”: The sentence is cumbersome; accordingly, kindly reformulate in order to make it clearer and more aiming.

Response: We appreciated with this suggestion. We have revised the sentences in the revised manuscript (Page 11, Lines 228-230).

  • Results, 3.3. KEGG pathway and GO enrichment analyses of the DEGs, Page 7: Kindly adjust as follow: “the genes” and “S. rugosoannulata”.

Response: Thank you very much! We have revised these words in the revised manuscript (Page 12, Lines 258).

  • Results, 3.3. KEGG pathway and GO enrichment analyses of the DEGs, Page 7: “The enriched… metabolism (Figure S1B)”: The sentence is badly written in standard English; accordingly, kindly reformulate it.

Response: We appreciated with this suggestion. We have reformulated this sentence in the revised manuscript (Page 12-13, Lines 263-266).

  • Results, 3.3. KEGG pathway and GO enrichment analyses of the DEGs, Page 8: Kindly replace “elevated” by “increased”.

Response: Thank you! We have revised it in the revised manuscript (Page 13, Line 283).

  • Results, 3.4. Analysis of nitrogen metabolic genes at the three nitrogen levels, Page 9: “Therefore… S. rugosoannulata”: Kindly avoid the first voice form of the sentence and adopt the impersonal form instead.

Response: Thank you very much! We have revised this sentence again in the impersonal form in the revised manuscript (Page 14, Lines 300-301).

  • Results, 3.4. Analysis of nitrogen metabolic genes at the three nitrogen levels, Page 9: Kindly adjust as follow: “and only one gene… was upregulated”.

Response: Thank you for the suggestion! We have revised it in the manuscript (Page 14, Line 307).

  • Results, 3.4. Analysis of nitrogen metabolic genes at the three nitrogen levels, Page 9: Kindly adjust as follow: “in response to”.

Response: According to the suggestion, we have revised it in the revised manuscript (Page 15, Line 310).

  • Results, 3.4. Analysis of nitrogen metabolic genes at the three nitrogen levels, Page 9: “We thus… nitrogen levels”: Kindly avoid the first voice form of the sentence and adopt the impersonal form instead.

Response: We appreciated with this suggestion. We have revised this sentence in the revised manuscript (Page 15, Lines 312-313).

  • Results, 3.4. Analysis of nitrogen metabolic genes at the three nitrogen levels, Page 9: Kindly adjust as follow: “no significant differences”.

Response: Thank you! We have revised it in the revised manuscript (Page 15, Line 315).

  • Results, 3.4. Analysis of nitrogen metabolic genes at the three nitrogen levels, Page 9: Kindly adjust as follow: “than the other two samples”.

Response: According to the suggestion, we have revised it in the revised manuscript (Page 15, Line 317).

  • Results, 3.4. Analysis of nitrogen metabolic genes at the three nitrogen levels, Page 9: “Besides… than A2”: The sentence is badly written in standard English; accordingly, kindly reformulate it

Response: We appreciated with this suggestion. We have revised these sentences in the revised manuscript (Page 15, Lines 317-318).

  • Results, 3.4. Analysis of nitrogen metabolic genes at the three nitrogen levels, Page 9: “The activity of GDH… metabolism enzymes”: Same recommendation as in the previous comments.

Response: We appreciated with this suggestion. We have revised this sentence in the revised manuscript (Page 15, Lines 319-322).

  • Results, 3.4. Analysis of nitrogen metabolic genes at the three nitrogen levels, Page 10: Kindly remove “significantly” before “differentially”.

Response: Thank you! We have removed the word “significantly” in the revised manuscript.

  • Results, 3.5. Analysis of carbon metabolic genes at the three nitrogen levels, Page 11: Kindy adjust the numbering of this paragraph as “3.5.” instead of “3.4.”

Response: According to the suggestion, we have numbered this paragraph of “Analysis of carbon metabolic genes at the three nitrogen levels” as “3.5” in the revised manuscript.

  • Results, 3.5. Analysis of carbon metabolic genes at the three nitrogen levels, Page 11: Kindly remove reference “[5]” from here.

Response: Thank you! We have removed the reference “[5]” in the revised manuscript.

  • Results, 3.5. Analysis of carbon metabolic genes at the three nitrogen levels, Page 11: “In this study… (Figure 5A)”: Kindly avoid the first voice form of the sentence and adopt the impersonal form instead.

Response: We appreciated with this suggestion. We have revised this sentence in the revised manuscript (Page 16, Lines 347-348).

  • Results, 3.5. Analysis of carbon metabolic genes at the three nitrogen levels, Page 11: Kindly adjust as follow: “no significant differences”.

Response: Thank you very much! We have revised it in the revised manuscript (Page 16, line 350).

  • Results, 3.5. Analysis of carbon metabolic genes at the three nitrogen levels, Page 11: “Furthermore… (Figure 5B)”: Kindly avoid the first voice form of the sentence and adopt the impersonal form instead.

Response: Thank you very much! We have revised this sentence in the revised manuscript (Page 17, Lines 353-355).

  • Results, 3.5. Analysis of carbon metabolic genes at the three nitrogen levels, Page 11: “During these genes… in A1”: The sentence is badly written in standard English; accordingly, kindly reformulate it.

Response: We appreciated with this suggestion. We have reformulated this sentence in the revised manuscript (Page 17, Lines 357-360).

  • Results, 3.5. Analysis of carbon metabolic genes at the three nitrogen levels, Page 12: “We also analyzed… (Figure 5B)”: Kindly avoid the first voice form of the sentence and adopt the impersonal form instead.

Response: We appreciated with this suggestion. We have revised this sentence in the revised manuscript (Page 17, Lines 369-371)

  • Results, 3.5. Analysis of carbon metabolic genes at the three nitrogen levels, Page 12: Kindly adjust as follow: “were upregulated”.

Response: Thank you! We have revised the word in the revised manuscript (Page 17, Line 372).

  • Results, 3.5. Analysis of carbon metabolic genes at the three nitrogen levels, Page 13: “Therefore… S. rugosoannulata”: Kindly avoid the first voice form of the sentence and adopt the impersonal form instead.

Response: Thank you very much! We have revised the sentence in the revised manuscript (Page 18, Lines 375-376).

  • Results, 3.5. Analysis of carbon metabolic genes at the three nitrogen levels, Page 13: Kindly adjust as follow: “were the highest in A2”.

Response: Thank you! We have revised it in the revised manuscript (Page 18, Line 384).

  • Results, 3.5. Analysis of carbon metabolic genes at the three nitrogen levels, Page 13: Kindly adjust as follow: “with the analyses of carbohydrate”

Response: Thank you! We have revised the writing in the revised manuscript (Page 18, Line 386).

  • Results, 3.5. Analysis of carbon metabolic genes at the three nitrogen levels, Page 13: “We also detected… [18-19]”: Kindly move this statement to the Discussion part. Also, kindly avoid the first voice form of the sentence and adopt the impersonal form instead.

Response: We appreciated with these suggestions. We have moved this statement to the discussion part (Page 22, Lines 465-467) and also revised the form of the sentence in the revised manuscript (Page 18, Lines 390-391).

  • Results, 3.5. Analysis of carbon metabolic genes at the three nitrogen levels, Page 13: “We found… (Figure 5G)”: Kindly avoid the first voice form of the sentence and adopt the impersonal form instead.

Response: Thank you very much! We have revised this sentence to avoid the first voice form in the revised manuscript (Page 18, Line 388-389).

  • Results, 3.6. Analysis of the MAPK signaling pathway at the three nitrogen levels, Page 13: Kindly adjust the numbering of this paragraph as “3.6.: instead of “3.5.”

Response: We appreciated with this suggestion. The result of “Analysis of the MAPK signaling pathway at the three nitrogen levels” was numbered as “3.6” in the revised manuscript.

  • Results, 3.6. Analysis of the MAPK signaling pathway at the three nitrogen levels, Page 13: “We found that… (DQGG008666)”: Kindly avoid the first voice form of the sentence and adopt the impersonal form instead. Moreover, the sentence is long and cumbersome; accordingly, kindly reformulate in order to make it more concise, clearer and more aiming.

Response: We appreciated with these suggestions. We have revised these sentences to avoid the first voice and to make it more concise, clearer and more aiming in the revised manuscript (Page 19, Lines 400-404).

  • Results, 3.6. Analysis of the MAPK signaling pathway at the three nitrogen levels, Page 13: “Besides… (Figure 6D)”: Kindly avoid the first voice form of the sentence and adopt the impersonal form instead.

Response: Thank you very much! We have revised the sentence in the revised manuscript (Page 19, Line 406-407).

  • Results, 3.7. Validation of transcriptomic data by qRT-PCR, Page 14: Kindly adjust the numbering of this paragraph as “3.7.” instead of “3.6.”

Response: Yes, this paragraph was numbered as “3.7” in the revised manuscript.

  • Results, 3.7. Validation of transcriptomic data by qRT-PCR, Page 14: Kindly adjust as follow: “similarly to”.

Response: Thank you! We have revised it in the revised manuscript (Page 20, Line 420).

  • Results, 3.7. Validation of transcriptomic data by qRT-PCR, Pages 14–15: “In addition… (Figure 7C)”: Kindly avoid the first voice form of the sentence and adopt the impersonal form instead.

Response: Thank you very much! We have revised it in the revised manuscript (Page 20, Line 421).

  • Results, 3.7. Validation of transcriptomic data by qRT-PCR, Page 15: Kindly adjust as follow: “similarly to”.

Response: Thank you! We have revised the word in the revised manuscript (Page 20, Line 422).

  • 4. Discussion, Page 16: “ rugosoannulata… in fungi [4]”: Kindly remove these sentences as you already mentioned that in the Introduction part.

Response: According to the suggestion, we have removed these sentences in the revised manuscript.

  • Discussion, Page 16: “In this study… quickly in A3”: Kindly avoid the first voice form of the sentence and adopt the impersonal form instead.

Response: Thank you very much! We have revised the sentence in the revised manuscript (Page 20, Lines 429-431).

  • Discussion, Page 16: “In addition… inhibited in A3”: Same recommendation as in the previous comment.

Response: Thank you very much! We have revised it in the revised manuscript (Page 20, Line 433).

  • Discussion, Page 16: Kindly adjust as follow: “nitrogen regulates”.

Response: Thank you! We have revised it in the revised manuscript (Page 21, line 440).

  • Discussion, Page 16: “In fungi… ACXs [15,18-19]”: It is better to replace these old references by newer ones as aforementioned.

Response: According to the suggestion, we have replaced these old references by newer ones in the revised manuscript (Page 21, Lines 440-450; Page 28, Lines 610-615; Page 29, Lines 619-626).

  • Discussion, Page 16: “In Pleurotus dryinus… [16]”: The sentence is badly written in standard English; accordingly, kindly reformulate it. Also, same recommendation as in the previous comment.

Response: We appreciated with this suggestion. We have revised these sentences in the revised manuscript (Page 21, Lines 442- 445).

  • Discussion, Page 16: “Kapich et al. [27] … ME-446”: Same point regarding the old references.

Response: Thank you very much! We have replaced the reference for new one in the revised manuscript (Page 21, Lines 445-448; Lines 30, Lines 650-653).

  • Discussion, Page 16: Kindly adjust as follow: “On the contrary”.

Response: Thank you again! We have revised it in the revised manuscript (Page 21, Lines 451-452).

  • Discussion, Page 16: “From these results… effects in fungi”: Kindly avoid the first voice form of the sentence and adopt the impersonal form instead. Moreover, kindly reformulate the sentence in a better language.

Response: We appreciated with this suggestion. We have revised this sentence in the revised manuscript (Page 22, Lines 462-464).

  • Discussion, Page 16: Kindly adjust as follow: “Similarly to the case of”.

Response: Thank you! We have revised it in the revised manuscript (Page 23, Line 464).

  • Discussion, Page 16: Kindly adjust as follow: “substrate of”.

Response: Thank you! We have revised it in the revised manuscript (Page 22, Line 468).

  • Discussion, Page 16: “Nitrogen metabolism… of nitrogen sources”: The sentence is badly written in standard English; accordingly, kindly reformulate it.

Response: Thank you very much! We have revised these sentences in the revised manuscript (Page 22, Lines 469-470).

  • Discussion, Page 17: “CEs represent… saccharides [37]”: Same point regarding the old references.

Response: Thank you very much! We have replaced this reference in the revised manuscript (Page 23, Lines 487-499).

  • Discussion, Page 17: “The CE10 family… levels in S. rugosoannulata”: Same recommendation as in the previous comment.

Response: Thank you! We have replaced this reference in the revised manuscript (Page 23, Lines 500-501).

  • 4. Discussion, Page 17: “During the KEGG… significantly enriched”: Kindly avoid the first voice form of the sentence and adopt the impersonal form instead.

Response: Thank you! We have revised it in the revised manuscript (Page 24, Lines 507-508).

  • Conclusions, Page 17: “In this study… than the A1 and A2”: Same recommendation as in the previous comment.

Response: Thank you very much! We have revised it in the revised manuscript (Page 24, Lines 521-522).

Thanks very much for your attention to our manuscript!

Best wishes!

Hui Chen

6, Jun, 2023

Round 2

Reviewer 2 Report

Comments to the Author:

Title: Effects of different nitrogen levels on the lignocellulolytic enzyme production and gene expression under straw-state cultivation in Stropharia rugosoannulata

Overview and general recommendation:

Authors made significant improvements to their manuscript and are well thanked. Only minor adjustments are still required and the manuscript can be considered suitable for publication.

My comments and queries for authors are detailed below in “Minor comments” section.

1.1.            Minor comments:

1-     Abstract: Page 2, lines 32–35: “The activities… in A1”: The sentence is long and cumbersome; accordingly, kindly reformulate in order to make it more concise, clearer and more aiming.

2-    1. Introduction: Page 4, lines 78–81: “Carbon is another… [11]”: I am sorry for not clarifying the suggested study in my first report. Here it is in a clearer form: https://doi.org/10.1088/1755-1315/1090/1/012020. It is the study of Weghemmi et al. (2022) on Pleurotus ostreatus.

3-  1. Introduction: Page 5, lines 107–110: “Overall… usage”: The sentence is a little bit long; accordingly, kindly reformulate in order to make it more concise, clearer and more aiming.

4-  2. Materials and Methods, 2.6. Determination of cellulase and hemicellulase activity: Page 9, line 178: Kindly adjust as follow: “enzymes”.

5-    3. Results, 3.1. Nitrogen levels regulated the mycelial growth of S. rugosoannulata: Page 10, line 217: Kindly add “respectively” after “1.37%”.

6-    3. Results, 3.4. Analysis of nitrogen metabolism genes at the three nitrogen levels: Page 15, line 313: Kindly adjust as follow: “were detected”.

7-      4. Discussion: Page 21, line 441: Kindly adjust as follow: “including those of”.

8-  4. Discussion: Page 21, line 448: Kindly adjust as follow: “Actinomycetes”.

9-      4. Discussion: Page 21, lines 451–452: Kindly adjust as follow: “In contrast”.

10-  4. Discussion: Page 22, line 464: Kindly adjust as follow: “to the case of”.

11- 4. Discussion: Page 22, lines 464–468: “Similarly… S. rugosoannulata”: The sentence is long and cumbersome; accordingly, kindly reformulate in order to make it more concise, clearer and more aiming.

12-  4. Discussion: Page 22, lines 475–476: Kindly remove “in S. rugosoannulata”.

13-     4. Discussion: Page 22, lines 501–502: Kindly adjust the sentence as follow: “suggesting that the latter are also upregulated…”

Only minor linguistic adjustments are needed and are outlined in the attached report.

Author Response

Dear Editor Aisa Safaya and Reviewers:

Thanks very much for your kind suggestion for my manuscript entitled “Effects of different nitrogen levels on the lignocellulolytic enzyme production and genes expression under straw-state cultivation in Stropharia rugosoannulata”(IJMS-2421182). We have carefully revised our manuscript according to your suggestions. The revised segment has been highlighted by red color in the revised manuscript. The point-to-point replies to these comments are listed below. We hope the revised manuscript will be suitable for publication.

Overview and general recommendation:

Authors made significant improvements to their manuscript and are well thanked. Only minor adjustments are still required and the manuscript can be considered suitable for publication.

My comments and queries for authors are detailed below in “Minor comments” section.

Response: We appreciate the positive comments about the manuscript. We consider these suggestions are all significant for our research work and paper writing. We have revised all the suggestion in the revised manuscript.

1.1.            Minor comments:

  • Abstract: Page 2, lines 32–35: “The activities… in A1”: The sentence is long and cumbersome; accordingly, kindly reformulate in order to make it more concise, clearer and more aiming.

Response: We appreciated with this suggestion. We have revised this sentence in the revised manuscript (Page 2, Lines 32-34).

  • Introduction: Page 4, lines 78–81: “Carbon is another… [11]”: I am sorry for not clarifying the suggested study in my first report. Here it is in a clearer form: https://doi.org/10.1088/1755-1315/1090/1/012020. It is the study of Weghemmi et al. (2022) on Pleurotus ostreatus.

Response: Thank you very much! We have added this reference in the revised manscript (Page 27, Lines 586-588).

  • Introduction: Page 5, lines 107–110: “Overall… usage”: The sentence is a little bit long; accordingly, kindly reformulate in order to make it more concise, clearer and more aiming.

Response: We appreciated with this suggestion. We have revised this sentence in the revised manscript (Page 5, Lines 107-109).

  • Materials and Methods, 2.6. Determination of cellulase and hemicellulase activity: Page 9, line 178: Kindly adjust as follow: “enzymes”.

Response: We appreciated with this suggestion. We have revised it in the revised manuscript (Page 8, Line 176).

  • Results, 3.1. Nitrogen levels regulated the mycelial growth of S. rugosoannulata: Page 10, line 217: Kindly add “respectively” after “1.37%”.

Response: Thank you very much! We have added “respectively” in the revised manuscript (Page 10, Line 216).

  • Results, 3.4. Analysis of nitrogen metabolism genes at the three nitrogen levels: Page 15, line 313: Kindly adjust as follow: “were detected”.

Response: Thank you very much! We have revised it in the revised manuscript (Page 15, Line 312).

  • Discussion: Page 21, line 441: Kindly adjust as follow: “including those of”.

Response: Thank you very much! We have revised it in the revised manuscript (page 21, Line 440).

  • Discussion: Page 21, line 448: Kindly adjust as follow: “Actinomycetes”.

Response: We appreciated with this suggestion. We have revised it in the revised manuscript (Page 21, Line 447).

  • Discussion: Page 21, lines 451–452: Kindly adjust as follow: “In contrast”.

Response: Thank you! We have revised it in the revised manuscript (Page 21, Line 450).

  • Discussion: Page 22, line 464: Kindly adjust as follow: “to the case of”.

Response: Thank you very much! We have revised it in the revised manuscript (Page 22, Line 463).

  • Discussion: Page 22, lines 464–468: “Similarly… S. rugosoannulata”: The sentence is long and cumbersome; accordingly, kindly reformulate in order to make it more concise, clearer and more aiming.

Response: We appreciated with this suggestion. We have revised this sentence in the revised manuscript (Page 22, Lines 463-467).

  • Discussion: Page 22, lines 475–476: Kindly remove “in S. rugosoannulata”.

Response: According to the suggestion, we have removed “in S. rugosoannulata” in the revised manuscript.

  • Discussion: Page 22, lines 501–502: Kindly adjust the sentence as follow: “suggesting that the latter are also upregulated…”

Response: Thank you very much! We have revised it in the revised manuscript (Page 23, Line 500).

Thanks very much for your attention to our manuscript!

Best wishes!

Hui Chen

8, Jun, 2023
